



# A machine learning approach to downscale EMEP4UK: analysis of UK ozone variability and trends

Lily Gouldsbrough[1], Ryan Hossaini[1,2], Emma Eastoe[3], Paul J. Young[1,4] and Massimo Vieno[5]

[1]Lancaster Environment Centre, Lancaster University, Lancaster, LA1 4YQ, UK
[2]Centre of Excellence in Environmental Data Science (CEEDS), Lancaster University, Lancaster, UK
[3]Mathematics and Statistics, Lancaster University, Lancaster, LA1 4YR, UK
[4]JBA Risk Management Limited, Broughton Park, Skipton, BD23 3FD, UK
[5]UK Centre for Ecology & Hydrology, Bush Estate, Penicuik, Midlothian EH26 0QB, UK

*Correspondence to*: Lily Gouldsbrough (lilygouldsbrough@outlook.com) and Ryan Hossaini
(r.hossaini@lancaster.ac.uk)

**Abstract.** High-resolution modelling of surface ozone is an essential step in the quantification of the impacts on health and ecosystems from historic and future concentrations. It also provides a principled way in which to extend analysis beyond measurement locations. Often, such modelling uses relatively coarse resolution chemistry transport models (CTMs), which exhibit biases when compared to measurements. EMEP4UK is a CTM that is
used extensively to inform UK air quality policy, including the effects on ozone from mitigation of its precursors. Our evaluation of EMEP4UK for the years 2001–2018 finds a high bias in reproducing daily maximum 8-hr average ozone (MDA8), due in part to the coarse spatial resolution. We present a machine learning downscaling methodology to downscale EMEP4UK ozone output from a 5×5 km to 1×1 km resolution using a gradient boosted tree. By addressing the high bias present in EMEP4UK, the downscaled surface better represents the measured
data, with a 128% improvement in $R^2$ and 37% reduction in RMSE. Our analysis of the downscaled surface shows a decreasing trend in annual and March–August mean MDA8 ozone for all regions of the UK between 2001–2018, differing from increasing measurement trends in some regions. We find the proportion of the UK which fails the government objective to have at most 10 exceedances of 100 μg/m³ per annum is 27% (2014–2018 average), compared to 99% from the unadjusted EMEP4UK model. A statistically significant trend in this
proportion of -2.19%/year is found from the downscaled surface only, highlighting the importance of bias correction in the assessment of policy metrics. Finally, we use the downscaling approach to examine the sensitivity of UK surface ozone to reductions in UK terrestrial $NO_X$ (i.e., $NO + NO_2$) emissions on a 1×1 km surface. Moderate $NO_X$ emission reductions with respect to present day (20% or 40%) increase both average and high-level ozone concentrations in large portions of the UK, whereas larger $NO_X$ reductions (80%) cause a similarly
wide-spread decrease in high-level ozone. In all three scenarios, very urban areas (i.e., major cities) are the most affected by increasing concentrations of ozone, emphasising the broader air quality challenges of $NO_X$ control.

## 1 Introduction

Ground level ozone is a harmful air pollutant that causes respiratory health issues, hospitalisation, and in severe cases, mortality (Ji et al., 2011; COMEAP, 2015; Nuvolone et al., 2018), with an estimated 254,000 deaths
globally in 2015 due to elevated ozone exposure (Cohen et al., 2017). Ozone is a secondary pollutant formed via chemical reactions of precursor pollutants – nitrogen oxides (NO and $NO_2$, known as $NO_X$) and volatile organic compounds (VOCs) – in the presence of sunlight. Because of the harmful consequences of increased ozone levels, air quality standards of varying degrees of stringency have been set in many countries for both ozone and its precursors. Compliance with, and the efficacy of, these standards is primarily assessed through analysis of surface
ozone measurements. Whilst useful, the empirical analysis of ozone measurements is limited in scope by the density and location of monitoring sites and length of data records (e.g., Lang, 2020). Mathematical models, either process-based air quality models or statistical, are therefore useful in further improving our understanding of ozone formation, long-term trends, and spatial variability, at spatial and temporal scales that measurements alone cannot match. However, it is well established that ozone concentrations vary spatially, seasonally and temporally
due to meteorological conditions and precursor availability and reactivity (Cooper et al., 2014; Pope et al., 2016), making it a particularly challenging pollutant to model.

Recent statistical analysis of UK surface ozone has centred on data from measurement stations (Diaz et al., 2020; Gouldsbrough et al., 2022). However, these stations are not spread equally across the UK, leaving substantial
portions of the country unmonitored (Finch and Palmer, 2020). In consequence, robust estimates of regional variability in ozone concentration and trends are unavailable, though it is known that both vary significantly across sites and site type. Taking the UK as a whole, a notable finding of the above studies is that the probability of the



occurrence of extreme ozone concentrations has reduced in recent decades, but reported trends are generally not statistically significant (e.g., Finch and Palmer, 2020). Chemistry transport models (CTMs), numerical models that simulate the various processes that affect pollutant concentrations (emissions, chemistry transport, deposition, etc.) are routinely used to produce spatial surfaces of ozone, and other air pollutants, with a far greater spatial coverage than can be achieved by the monitoring stations. Still, these are run on a grid, often at a coarse resolution with respect to what is optimal for exposure assessment, and thus may not capture local-scale behaviour. High resolution multi-annual CTM simulations are also computationally expensive and, like all process models, subject to a degree of bias (e.g., Liu et al., 2022). An alternative to high-resolution process modelling is to develop a statistical model to interpolate the ozone measurements (Wong et al., 2004; Hooyberghs et al., 2006). However, given the complexity of the processes that underpin ozone production, downscaling approaches that use measured data, where it is available, to remove local-scale bias from the numerical model output are attractive.

Data-driven downscaling methods are used to model complex physical systems by combining information from ground observations or satellites with information from process-based numerical models. Previous work to downscale numerical model surface ozone data includes dynamical, statistical and machine learning downscaling. Dynamical downscaling uses high resolution regional simulations to extrapolate the effects of large scale processes to local scales, and has been applied in the US (Nolte et al., 2021; Sun et al., 2015; Trail et al., 2013) and Belgium (Lauwaet et al., 2013). Statistical downscaling of surface ozone has been performed using regression (Bravo et al., 2016; Gauthier-Manuel et al., 2022; Guillas et al., 2008), fitted empirical orthogonal functions (Alkuwari et al., 2013), and a spectral method (Reich et al., 2014). Machine learning (ML) models have also been used to downscale surface ozone. For example, a Bayesian ensemble machine learning model that integrates 13 learning algorithms has been used to create a census tract level daily maximum 8-hr average ozone (MDA8) ozone surface for the US, demonstrated for 2011 (Ren et al., 2022). Global downscaled ozone surfaces have also been created using ML models, including a Bayesian neural network model to create a $10\times10$ km ozone surface for 1990–2019 (Sun et al., 2022), and a random forest model to create a $0.1°\times0.1°$ average ozone surface for 2010–2014 (Betancourt et al., 2022). To our knowledge, no studies have applied downscaling specifically to UK ozone.

In this paper, we develop and evaluate a novel ML-based methodology for downscaling the EMEP4UK CTM from a $5\times5$ km to $1\times1$ km resolution. The CTM is developed as the UK regional application of the European Monitoring and Evaluation Program (EMEP) and has been widely used to study UK air quality and to inform policy decisions (e.g., Vieno et al., 2016). Previous EMEP4UK evaluation shows the model generally performs well at reproducing observations of a range of pollutants, though notably ozone exhibits a non-negligible positive bias at almost all sites. The bias is larger at urban background locations, possibly reflecting a dilution of local $NO_X$ emissions on the $5\times5$ km grid and, thus, insufficient $NO_X$ titration (Lin et al., 2017). We apply the downscaled model to study UK surface ozone over an 18-year period (2001–2018), focussing on (1) quantifying regional variability in ozone, trends, and policy-related metrics; (2) comparing conclusions drawn from the downscaled surface relative to those from the unadjusted CTM and measurements alone; and (3) exploring the sensitivity of UK ozone concentrations to reductions in $NO_X$. Of the many ML tools available, we use a gradient boosted tree (GBT) (Friedman, 2001). Our choice is primarily driven by the fact that the GBT model can learn non-linear relationships from highly dimensional datasets. Thus, the GBT model allows us to use several measurement stations and covariates in the downscaling model, and thereby model ozone as the product of multiple highly non-linear and interacting systems. Moreover, GBTs have been used to downscaled numerical model surface ozone for China (Hu et al., 2022; Liu et al., 2020), and were found useful in predicting surface level ozone during wildfires in California (Watson et al., 2019).

This paper is structured as follows. Section 2 outlines the data used in the analysis and the features included with the ML model. Section 3 describes the downscaling methodology, including an evaluation of the downscaled surface. In Section 4 we analyse the downscale surface, focussing on regional ozone variability across the UK in recent years (Section 4.1), on longer-term trends and interannual variability (4.2), and on UK ozone-$NO_X$ sensitivity (4.3). Across these results we compare the conclusions drawn from the new downscaled surface to those from the unadjusted EMEP4UK output and measurements alone. Finally, in Section 5 we present our conclusions and some brief recommendations for future research.

## 2 Data

Accurate downscaling requires not just the measured and modelled ozone data, but also information on variables which affect the net production or transport of ozone at the Earth's surface. We make use of information on the meteorology, climate and geophysical characteristics at a given location. For consistency with ML terminology, we refer to the individual variables as 'input features', a concise summary of which can be found in **Table 1**. All data sets cover the period 2001–2018 (inclusive).



### 2.1 Modelled ozone from EMEP4UK

EMEP4UK is a UK focussed version of the EMEP MSC-W model (https://www.emep.int), an Eulerian CTM used to assess concentrations and deposition of various air pollutants across Europe (Simpson et al., 2012).
Various studies related to UK ozone have been performed using EMEP4UK: quantifying the burden of heat and ozone on mortality (Doherty et al., 2009), modelling ozone during the 2003 heatwave (Vieno et al., 2010), modelling the effect of climate change on ozone health impacts (Vardoulakis and Heaviside, 2012), quantifying the socioeconomic and urban-rural differentials in exposure (Milojevic et al., 2017), and modelling air pollution exposure in relation to workplace mobility (Liška, 2021). The EMEP4UK model is also used to inform policy decisions concerning air quality: the extent to which UK source abatement measures can mitigate UK particulate matter concentrations (AQEG et al., 2013; Carnell et al., 2019); the impact of reductions in UK anthropogenic emissions on various pollutants (Vieno et al., 2016); the impacts of climate change and mitigation options for agriculture, forestry, land use and waste sectors (SRUC, 2017); the effect of changes in vegetation coverage on air pollution (EIDC, 2021); and to quantify the spatial variation in average ozone across the UK, including the calculation of population-weighted ozone exposure during workdays, long term exposure, and the implication of a 2030 emissions scenario on surface ozone concentrations (AQEG, 2021).

Previous evaluation of the EMEP4UK model quantified its performance in the reproduction of the 10-year mean measured ozone concentrations for 2001–2010. Considering 17 rural and 30 urban sites, $R^2$ values of 0.21 (0.81 when removing one erroneous rural site) and 0.73, respectively, were obtained (Lin et al., 2017). A positive model bias for ozone at urban background sites is due to the dilution of urban $NO_X$ emissions at the model 5×5 km resolution, meaning that EMEP4UK insufficiently captures the urban $NO_X$ titration of ozone. In our preliminary investigations, we found that the original EMEP4UK output fails to capture the daily behaviour of ozone in the larger sample of 198 measurement stations used in this analysis, with a cross-year mean $R^2$ of 0.32, and cross-year mean RMSE of 19.4 $\mu g/m^3$. Therefore, we determined it was necessary to develop a methodology with which to downscale the EMEP4UK model output to a higher resolution spatial grid and to address the above bias.

The dataset we wished to downscale is a 5×5 km gridded dataset of hourly surface ozone concentrations from EMEP4UK covering the period 2001–2018. To produce this ozone field, the offline CTM was run with meteorology from the Weather Research and Forecast (WRF) model version 3.7.1 (Skamarock et al., 2008) between 2001–2017 and WRF4.1 (Skamarock et al., 2019) for 2018. The WRF simulation in this work assimilates data from the numerical weather prediction model meteorological reanalysis of the US National Center for Environmental Prediction (NCEP)/National Center for Atmospheric Research (NCAR) Global Forecast System (GFS) (National Centers for Environmental Prediction, 2000). MDA8 ozone concentrations were calculated from the original hourly model output and linear interpolation was used to convert the 5×5 km field to a 1×1 km grid over the entire study period (2001–2018).

In addition to the output from the main transient model run described above, a series of additional model experiments were performed for the year 2018 to explore the sensitivity of UK ozone to $NO_X$ reductions. These included a 2018 reference run with terrestrial UK $NO_2$ emissions of 743 Gg/yr and three otherwise identical runs with $NO_2$ emissions reduced by 20% (594 Gg/yr $NO_2$), 40% (446 Gg/yr), and 80% (148 Gg/yr). These reductions do not correspond to any specific future scenarios and are designed solely as a sensitivity analysis on the basis of the ongoing long-term decline in UK $NO_X$ emissions. These emissions fell by 76% between 1970 (2920 Gg/yr) and 2020 (702 Gg/yr) (Defra, 2021a), with the expectation of further reductions in the future.

### 2.2 Meteorological variables

Surface ozone levels are strongly influenced by local and synoptic weather conditions (Pope et al., 2016) and meteorological variables are thus common input features in ML studies of ozone. The WRF model is a weather prediction system designed for atmospheric forecasting (Grell et al., 2005). For this study, WRF version 3.7.1 meteorological and terrain variables for the years 2001–2018 were collected on the same 5×5 km grid as the EMEP4UK model, and then also linearly interpolated to 1×1 km. Previous work found daily maximum temperature, relative humidity, thermal surface radiation and wind speed to be important drivers of MDA8 ozone in Europe (Otero et al., 2016). Thus, we include these (and other similarly relevant) meteorological variables in our ML model (**Table 1**).

### 2.3 Distance variables

Due to the link between $NO_X$ and ozone, distance to the nearest road is a key explanatory variable (Granier and Brasseur, 2003). Distances to five road types from major to minor roads (Meijer et al., 2018) were calculated at each ozone measurement station for the calibration of the model, and on a 1×1 km grid for the predicted downscaled surface. Similarly, distance to coast from a shapefile of the UK coastline (Natural Earth: https://www.naturalearthdata.com) was used to account for the increase of ozone concentrations in coastal areas



(Entwistle et al., 1997). Note that we do not include $NO_X$ itself, or any other precursor pollutants, directly in the model. The primary reason for this is that the EMEP4UK output for such chemicals is likely to be biased, and we wish to avoid this bias propagating into the downscaled ozone field. Instead, the distance to road variable acts as a proxy for $NO_X$ concentration, which is reasonable given the importance of road transport $NO_X$ emissions in the UK. To lessen the bias of the downscaling model towards the relatively dense measurement network in London, an indicator variable was included to delineate between inside London and outside London, with London defined using a bounding box from 0.489°W–0.236°E and 51.28–51.686°N.


**2.4 Ozone monitoring network data**


Surface ozone measurements for the years 2001–2018 were obtained from the Automatic Urban and Rural Network (AURN: https://uk-air.defra.gov.uk, 108 sites), Kings College London network (KCL: https://www.londonair.org.uk, 68 sites), Air Quality England network (AQE: https://www.airqualityengland.co.uk, 12 sites), Welsh Air Quality Network (WAQN: https://airquality.gov.wales, 9 sites) and Scottish Air Quality Network (SAQN: https://www.scottishairquality.scot, 1 site). These measurements are essential for the calibration and evaluation of the downscaling model. MDA8 ozone concentrations were calculated at each of the 198 measurement sites; for locations, see Figure A1. There are differences in the observation period across the sites, but all sites had a minimum of 3 years data.


In our subsequent analysis, we explored regional variations in ozone concentration and trends. We considered 12 UK regions (see Figure A2), the spatial definitions of which are taken from the Level 1 Nomenclature of Territorial Units for Statistics (Office for National Statistics, 2018). Estimation of site-wise trends, including the magnitude and significance, may be sensitive to the chosen statistical technique. Approaches used in previous studies include use of the non-parametric Theil-Sen method applied to deseasonalised monthly mean ozone time series (AQEG, 2021) and least square fits to annual mean data (Finch and Palmer, 2020). In this study, we calculated trends using ordinary least squares on the yearly averages (and seasonal averages, 90th percentiles, 10th percentiles) of MDA8 ozone. All trends are calculated using all available data from each region.



**Table 1: Input features to the ML model.**

| Input features | Source | Abbreviation in source | Resolution |
|---|---|---|---|
| EMEP4UK surface ozone | EMEP4UK | O3 | 1×1 km |
| Latitude | Measurements | -- | -- |
| Longitude | Measurements | -- | -- |
| Daily maximum 2m temperature | WRF | T2 | 1×1 km |
| Daily minimum 2m temperature | WRF | T2 | 1×1 km |
| Daily mean surface pressure | WRF | PSFC | 1×1 km |
| Downward short-wave flux at ground surface | WRF | SWDOWN | 1×1 km |
| Daily mean planet boundary layer height | WRF | PBLH | 1×1 km |
| Daily mean surface vapor | WRF | QVAPOR | 1×1 km |
| Daily mean x component of wind | WRF | U10 | 1×1 km |
| Daily mean y component of wind | WRF | V10 | 1×1 km |
| Terrain height | WRF | HGT | 1×1 km |
| Distance to highways | GRIP | 1 | Vector |
| Distance to primary roads | GRIP | 2 | Vector |
| Distance to secondary roads | GRIP | 3 | Vector |
| Distance to tertiary roads | GRIP | 4 | Vector |
| Distance to local roads | GRIP | 5 | Vector |
| Distance to coast | Natural Earth | -- | Vector |
| Year | EMEP4UK | -- | -- |
| Month | EMEP4UK | -- | -- |
| Date (as integer) | EMEP4UK | -- | -- |
| London (or not) | Bounding box | -- | -- |


**3 Downscaling methodology**

Our goal is to produce a gridded downscaled surface ozone product which better represents the stochastic *behaviour* of the measurements than the original EMEP4UK output alone. The downscaling approach consists of five steps. First, the 5×5 km gridded model ozone surface is linearly interpolated to a 1×1 km resolution. Second, a matched data set of modelled ozone is selected from the 1×1 km EMEP4UK surface by selecting, for each




measurement station, the nearest grid cell. Third, a machine learning model is used to perform bias correction on the modelled ozone data. Fourth, the performance of the bias correction is evaluated at the measurement locations, by a comparison of the predicted ozone with the observed measurements. Steps three and four are iterated until no further improvements in the predictive capability of the model can be seen. Finally, the trained machine
learning model is used to predict MDA8 surface ozone on a 1×1 km resolution grid for the UK. **Figure 1** shows an example of the resulting downscaled surface. This surface consists of 234,187 cells, compared to the 10,941 cells of the original EMEP4UK surface. The increased resolution in the downscaled surface leads to greater local-level detail, resulting in improved inference on the probabilistic behaviour of the ozone surface which we demonstrate in our subsequent analysis. Further details on the specific machine learning model, and how it was
tuned, are as follows.

**Figure 1: (a) An example of the downscaled MDA8 (µg/m³) surface (1×1 km resolution) and (b) original EMEP4UK surface (right, 5×5 km) for 01-01-2008.**


### 3.1 Machine learning model

A gradient boosting tree (GBT) is an iterative, supervised, machine learning model, consisting of a parameterized ensemble of decision trees (Friedman, 2001). These decision trees are trained sequentially; each additional tree minimizes the prediction error from the previous tree using gradient descent. As a supervised learning algorithm,
training the model requires both a training data set and a predefined objective function, the latter consisting of a loss function and a regularization term. The training data set is the subset of the full set of data from which the best fitting model is found. The loss function and regularization term quantify the quality of model fit given the complexity of the model. The error for each decision tree is calculated from the loss function (Friedman, 2001), which measures how well the model predicts the training data, whilst the regularization term penalises against
model complexity to prevent overfitting. The fitting algorithm ends when either a predetermined number of trees have been fitted, the loss function falls below a predetermined threshold, or the addition of more trees provides no significant improvement to the model fit. The latter criterion is determined by an external validation data set. The final model is then the summation over the entire ensemble.

Several characteristics of GBTs make them suitable for downscaling: they can capture non-linear relationships between variables far more effectively than competitor approaches (e.g., statistical regression models) and they are both computationally efficient and scalable, i.e., are suitable for large datasets (Chen and Guestrin, 2016). The specific GBT implementation used for this analysis is XGBoost, a highly optimized Python package (Chen and Guestrin, 2016). Since the measurement ozone data is long tailed, we chose a gamma regression for objective and
evaluation functions of the GBT model; gamma regression is suited to modelling continuous, non-negative and long tailed data. Like many ML models, the fitting process used to train the GBTs prioritises the fit of the mean behaviour at the expense of characterising the tails (i.e., largest and smallest observations). To reduce this inequality and reduce the mean bias, the tails of the measurement data – data above (and below) high (and low) concentrations of ozone – were oversampled. Resampling is a common approach to rebalance the distribution of
training data for a ML model when the goal is to forecast rare values of the target variable (Torgo et al., 2015).



Lastly, as with any machine learning analysis, we require a balance between the fit of the model on the training data and with the ability to apply the model to unseen data, i.e., making sure that the model is not overfit or underfit to the training data. To get satisfactory results, the hyperparameters of the XGBoost model needed substantial tuning. Initial hyperparameters were found using Hyperopt, a Bayesian optimization package (Bergstra et al., 2013). Subsequent fine tuning was performed manually until no further improvement could be found in the cross-validation (CV) tests.

### 3.2 Evaluation of downscaling

### 3.2.1 Predicting ozone at measurement locations

Evaluation of the downscaling model used CV to assess the prediction of MDA8 ozone across multiple measurement locations. CV requires the model to be trained on a random subsample of the whole dataset, and the resulting model then used to predict the remaining, and previously unseen, data. The first CV test is to split the data into two random samples, selected from the entire dataset: 70% of the data is used to train the model and the remaining 30% is used for evaluation. **Table 2** shows the annual $R^2$ and RMSE for the predictions combined across all sites. We found a good agreement between predicted ozone and measurement ozone, with a cross-year mean $R^2$ of 0.80 and RMSE of 10.61 μg/m$^3$ for 2001–2018, and no evidence of substantial between-year variation in performance.

The second CV test assesses predictive performance at locations which are completely excluded from the model training. To do this, 10-fold CV was applied, with the measurement stations randomly split into 10 groups and each group used to evaluate the model trained on the remaining 9 groups. Again, we found a good agreement between predicted and measurement ozone, with a cross-year mean $R^2$ of 0.70, as seen in **Table 2**.

**Table 2: $R^2$ and RMSE (μg/m$^3$) results of predicted MDA8 ozone vs MDA8 measurements for the two cross-validation tests: 70/30 train/test split and 10-fold CV.**

| Year | 70/30 train/test split | | 10-fold CV | |
|------|-----|------|-----|------|
| | $R^2$ | RMSE | $R^2$ | RMSE |
| 2001 | 0.82 | 11.33 | 0.71 | 14.09 |
| 2002 | 0.79 | 11.36 | 0.70 | 13.72 |
| 2003 | 0.82 | 12.88 | 0.74 | 15.52 |
| 2004 | 0.79 | 11.42 | 0.69 | 13.82 |
| 2005 | 0.77 | 11.50 | 0.66 | 13.98 |
| 2006 | 0.84 | 11.66 | 0.74 | 14.72 |
| 2007 | 0.82 | 10.09 | 0.72 | 12.52 |
| 2008 | 0.83 | 10.56 | 0.74 | 12.89 |
| 2009 | 0.81 | 10.23 | 0.71 | 12.50 |
| 2010 | 0.79 | 10.43 | 0.68 | 12.87 |
| 2011 | 0.79 | 10.43 | 0.67 | 13.06 |
| 2012 | 0.79 | 10.18 | 0.68 | 12.61 |
| 2013 | 0.81 | 10.09 | 0.71 | 12.52 |
| 2014 | 0.77 | 10.22 | 0.65 | 12.56 |
| 2015 | 0.77 | 9.66 | 0.65 | 12.05 |
| 2016 | 0.80 | 9.88 | 0.73 | 11.63 |
| 2017 | 0.81 | 9.29 | 0.69 | 11.80 |
| 2018 | 0.83 | 9.81 | 0.73 | 12.42 |
| **Mean** | **0.80** | **10.61** | **0.70** | **13.07** |

### 3.2.2 Downscaled surface vs measurements

Having verified the accuracy of the ML downscaling model, we created the downscaled surface by training the ML model on all measurement data. To compare the downscaled surface to both measurements and the unadjusted EMEP4UK surface, the data from the cell nearest to each measurement station was extracted. Due to the different grid specifications, the matched cells were not concentric, but they are very close. **Table 3** shows the $R^2$ for each year for the downscaled surface and original EMEP4UK surface when compared to the measurement data. The cross-year mean $R^2$ of the downscaled surface is 0.73, 128% higher than the equivalent for the unadjusted EMEP4UK, at 0.32. Notably, the extremely low $R^2$ values for 2014–2016 in the latter are significantly improved in the downscaled surface, e.g., 0.06 vs 0.73 in 2016. Notable improvement is also seen in 2003, the best performing year for EMEP4UK, with a downscaled $R^2$ of 0.81 compared with 0.61 for unadjusted EMEP4UK.



There is also a 37% reduction in the cross-year mean RMSE for the downscaled surface compared with the unadjusted product: 12.26 µg/m³ vs 19.40 µg/m³.


**Table 3: R² and RMSE (µg/m³) results for downscaled ozone vs measurements and EMEP4UK ozone vs measurements.**

| Year | Downscaled R² | Downscaled RMSE | EMEP4UK R² | EMEP4UK RMSE |
|------|------|------|------|------|
| 2001 | 0.79 | 12.10 | 0.45 | 19.52 |
| 2002 | 0.77 | 12.05 | 0.30 | 20.91 |
| 2003 | 0.81 | 13.10 | 0.61 | 19.05 |
| 2004 | 0.76 | 12.25 | 0.47 | 18.01 |
| 2005 | 0.73 | 12.54 | 0.36 | 19.22 |
| 2006 | 0.80 | 13.13 | 0.54 | 19.81 |
| 2007 | 0.76 | 11.64 | 0.41 | 18.08 |
| 2008 | 0.78 | 11.97 | 0.44 | 19.07 |
| 2009 | 0.74 | 12.05 | 0.31 | 19.45 |
| 2010 | 0.70 | 12.65 | 0.29 | 19.40 |
| 2011 | 0.69 | 12.62 | 0.29 | 19.17 |
| 2012 | 0.71 | 12.02 | 0.29 | 18.88 |
| 2013 | 0.72 | 12.24 | 0.28 | 19.58 |
| 2014 | 0.65 | 12.73 | 0.08 | 20.54 |
| 2015 | 0.64 | 12.20 | 0.07 | 19.66 |
| 2016 | 0.73 | 11.46 | 0.06 | 21.52 |
| 2017 | 0.70 | 11.70 | 0.14 | 19.73 |
| 2018 | 0.74 | 12.15 | 0.45 | 17.62 |
| **Mean** | **0.73** | **12.26** | **0.32** | **19.40** |

Further differences in the capabilities of the two gridded products to represent surface ozone measurements are shown in **Figure 2**. A considerable reduction in noise can be seen in the scatter density plots of the downscaled
surface vs measurements (panel a), which has a stronger linear signal and less scatter in comparison to the equivalent plot for EMEP4UK (panel b). This indicates that the unadjusted EMEP4UK surface is a less accurate representation of the measurement data. The bias in the stochastic behaviour of the unadjusted EMEP4UK output is evident when comparing the percentiles of the measurement data to the percentiles of the (i) unadjusted and (ii) downscaled EMEP4UK (panel c); the percentiles for the unadjusted output are consistently higher than those of
the measurements, whilst those of the downscaled data are almost identical to those of the measurements. Similarly, we see a considerable shift in the density of the EMEP4UK percentiles compared to that of the measurement data (panel d); again, there is a far smaller discrepancy between the densities of the measurements and the downscaled data.



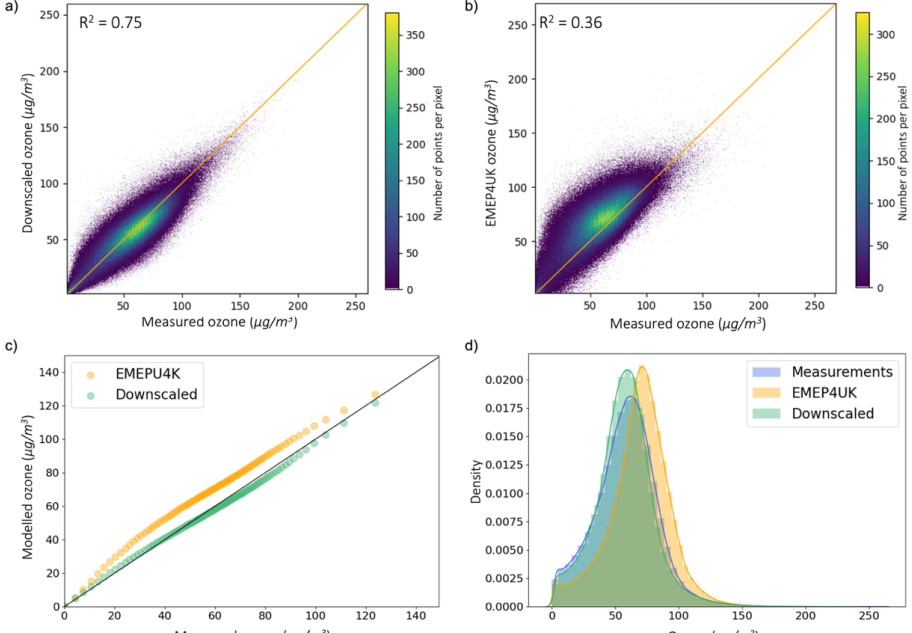

**Figure 2: a) scatter density plot of downscaled surface vs measurement ozone; b) scatter density plot of EMEP4UK surface vs measurement ozone; c) percentile-percentile plot comparing ordered percentiles of downscaled vs measurement ozone (green) and EMEP4UK vs measurement ozone (orange); and d) density plots of measurement ozone (blue), corresponding downscaled surface ozone (green) and corresponding EMEP4UK surface ozone (orange).**

### 3.3 Feature importance

Complex ensemble models, of which GBTs are an example, can be difficult to interpret. We make use of Shapley Additive exPlanations (SHAP) to quantify the importance of the input features to the trained GBT, and hence their importance to the predictive process, and to show that these features are consistent with what would be expected given our understanding of the generating mechanisms of surface ozone. The Shapley values (Lundberg and Lee, 2017) are one measure of feature importance that have been used previously to understand the relationship

between input features and ozone in ML studies (e.g., Liu et al., 2022). It is important to note that SHAP values cannot be interpreted as correlation coefficients; positive SHAP values can co-occur with either high (red) or low (blue) values of a feature, and similarly for negative SHAP values. This means that high values of a given feature may result in low or high ozone levels.

**Figure 3** shows the feature importance (as SHAP values) for the final GBT model trained on all data, where negative SHAP values result in lower predictions and positive ones to higher predictions. Unsurprisingly, EMEP4UK ozone is the most important feature in predicting the measured ozone, followed by daily maximum 2m temperature, date (as an integer), month, and distance to road type 1 (i.e., motorways). Lower concentrations of EMEP4UK ozone have a greater impact on the GBT model output than high values, signifying that lower

EMEP4UK ozone concentrations better represent the behaviour of measurement ozone than higher concentrations. Daily maximum 2m temperature is the most important meteorological feature, reflecting the well-established observed temperature/MDA8 relationship (e.g., Gouldsbrough et al., 2022) that is likely underpinned by several processes (Sun et al., 2017; Romer et al., 2018; Porter and Heald, 2019). Lower temperatures decreased the model prediction while higher temperatures increase the prediction. The high importance of the temporal

features (date and month) indicates that seasonality and long-term trends of measurement ozone are not wholly captured in EMEP4UK. Road type 1 is the most important of the road types, and the fifth most important feature overall, reflecting the strong link between vehicles, $NO_X$, and ozone. Type 1 roads typically have a higher traffic volume and considerably higher $NO_X$ concentrations than background locations, due to higher driving speeds and numbers of heavy good vehicles (Mann, 1997).




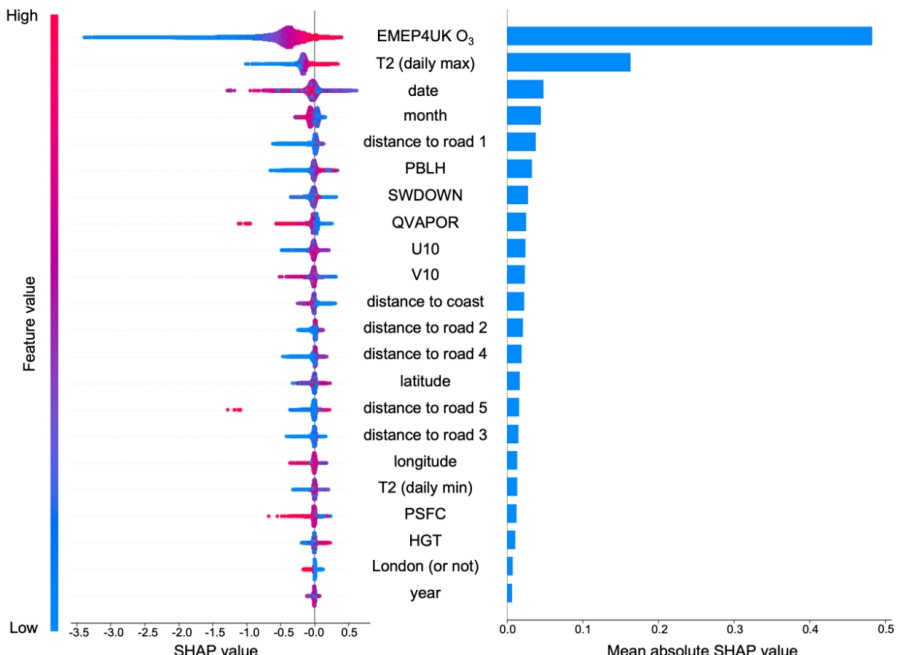


**Figure 3: Feature importance in the GBT model ordered from most important (top), to least important (bottom). Local SHAP values (left) show the model impact of each feature based on feature value, negative SHAP value results in a lower ozone prediction, and positive SHAP value results in a higher ozone prediction. Mean absolute SHAP values (right) show the overall impact of each feature on model output.**

### 4 Results - analysis of the downscaled surface


We perform three analyses of our downscaled ozone surface: recent years (2014–2018), time trends (2001–2018), and heatwave years (2003, 2006 and 2018). In all cases, we compare the behaviour of four characteristics – annual mean, March–August mean, and the annual 10$^{th}$ and 90$^{th}$ percentiles – across the measurement stations and the EMEP4UK and downscaled surfaces.


#### 4.1 Recent years analysis (2014–2018)


We examine the years 2014–2018 as these are the most recent years in the dataset. The five-year period accommodates the interannual variability in ozone concentrations, resulting in a broader overview of ozone behaviour. In the UK, elevated ozone mostly occurs in spring/summer (March–August) during anticyclonic conditions when slow moving air masses from mainland Europe contribute to increased accumulation of precursor emissions and increased rates of photochemical ozone production (AQEG, 2021). **Figure 4** shows the 2014–2018 annual (i.e., all months) and March–August (only) mean MDA8 ozone for each region and each data product. Recall that the measurement means are based on a limited and varying number of monitoring sites within each region (Figure A1). Regional means, both annual and March–August, are consistently higher for the original


EMEP4UK surface compared to the downscaled surface and measurements reflecting a high bias in the unadjusted CTM. See also Tables A1 and A2 which contain a summary of the data plotted in **Figure 4**. The all-region annual mean MDA8 from the downscaled surface (~62 µg/m$^3$) and measurements (~61 µg/m$^3$) are in close agreement, while the original EMEP4UK surface (~76 µg/m$^3$) is significantly larger (Table A1). A similar pattern of agreement is found for the March–August means (Table A2). One region where the downscaled surface and


measurements differ considerably is London. Here, the annual mean MDA8 for the downscaled surface and measurements are 57 µg/m$^3$ and 49 µg/m$^3$, respectively (and 68 µg/m$^3$ versus 58 µg/m$^3$ for the March–August mean). The high proportion of urban measurement sites in London (i.e., sampling more NO$_X$ titration), in contrast to the more varied site type sampling of the gridded downscaled surface, likely contributes to the higher MDA8 ozone in the downscaled surface.




While numerous studies have reported differences in UK ozone across different site types based on measurement analysis (Diaz et al., 2020; Finch and Palmer, 2020), there has been less focus on characterising regional variability. Considering the regional averages, analysis of the downscaled surface reveals a relatively modest amount of inter-region variability. For instance, the difference between the highest and lowest regional mean is 10 µg/m³ for the annual mean and 13 µg/m³ for the March–August mean. Similarly, across all regions the relative standard deviation is less than 6%. Variability in the regional means based on the original EMEP4UK surface is smaller (<1%). Additionally, we find that the region with the highest annual mean MDA8 ozone differs across all three data sets: Southwest England at 66 µg/m³ in the downscaled surface, Wales at 79 µg/m³ in the original EMEP4UK surface, and Southeast England at 67 µg/m³ in the measurements. Comparatively, the region with the highest March–August mean is Southeast England at 75 µg/m³ for both the downscaled surface and measurement data, whilst the highest March–August mean for EMEP4UK is East England at 88 µg/m³.

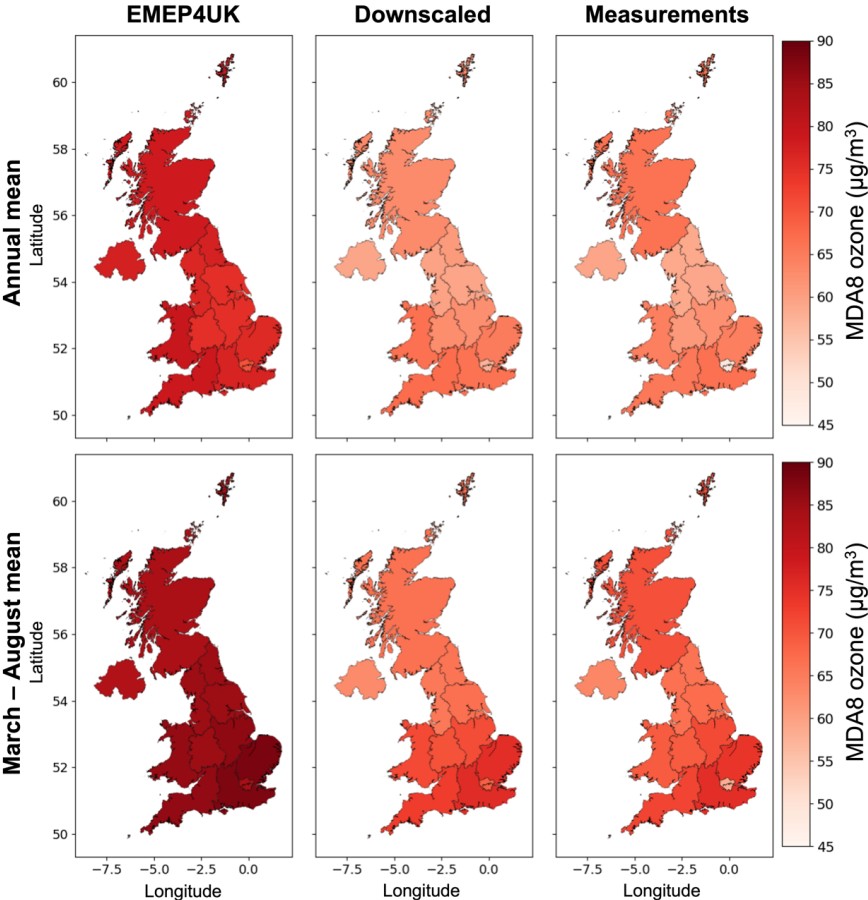

**Figure 4: Comparing regional averages of EMEP4UK (left column), downscaled (middle column), and measurement (right column) MDA8 ozone (µg/m³) for 2014–2018 annual mean (top row) and March–August mean (bottom row).**

**Figure 5** shows the 2014–2018 regional average 90th and 10th percentiles of MDA8 ozone. The EMEP4UK 90th and 10th percentiles are higher for all regions compared to the downscaled surface and measurements, further demonstrating the high ozone bias present in the EMEP4UK surface. Southeast England has the highest regional 90th percentile MDA8 ozone concentration in both the downscaled (88 µg/m³) and EMEP4UK (99 µg/m³,) surfaces, while Southwest England has the highest 90th percentile in the measurements at 90 µg/m³. The most noticeable difference between the three datasets is the 90th percentile estimate for Scotland: 79 µg/m³ for the downscaled surface, 95 µg/m³ for the EMEP4UK surface, and 86 µg/m³ for the measurements. Wales has the highest 10th percentile MDA8 ozone concentration in the EMEP4UK and downscaled surfaces, 62 µg/m³ and 50



μg/m³ respectively, while having only the third highest 10th percentile in the measurements, at 40 μg/m³. The highest regional 10th percentile in the measurement data is 45 μg/m³ for Scotland. The 10th percentiles for Scotland in the EMEP4UK and downscaled surfaces are higher still: 61 μg/m³ and 48 μg/m³, respectively. The relatively high 10th percentile in Scotland is likely due to the low regional NOₓ emissions, as ozone in Northern Scotland reflects hemispheric background concentrations instead of the photochemical generated concentrations (Entwistle et al., 1997). The inter-region variation in 90th percentile is 12 μg/m³ in the downscaled surface, whereas the inter-

region variation in the 10th percentile is considerably higher at 21 μg/m³, due to the particularly low 10th percentile ozone concentration in London of 29 μg/m³. See Table A3 and Table A4 for regional point estimates and confidence intervals and point estimates.

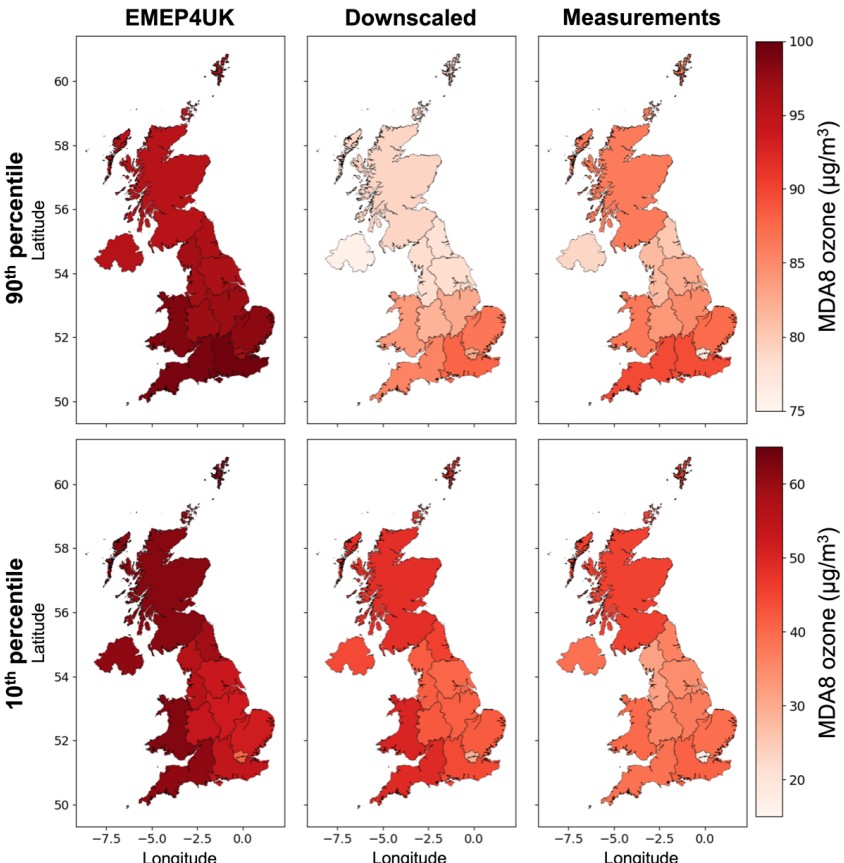

**Figure 5: Comparing regional averages of EMEP4UK (left column), downscaled (middle column), measurement (right column) MDA8 ozone for 2014–2018 90th percentile (top row) and 10th percentile (bottom row).**

Reports from Defra highlight that very few UK regions currently meet the UK government long-term ozone objective of MDA8 to not exceed 100 μg/m³ more than 10 times in a year. Similarly, the EU's long-term objective

(no MDA8 exceedances of 120 μg/m³) is routinely breached in most areas. A summary of the guidelines against which the UK reports is given in Table 1 of (Defra, 2021b). For the above assessments, which are based on measurements from the AURN network, if one location (monitoring site) within a region is in breach of the objective, that region is deemed non-compliant. For example, in 2021 no regions outside of Scotland met the EU objective (Defra, 2022). Our downscaled surface provides an additional perspective on adherence that is not

possible to obtain from a relatively sparse monitoring network alone or from a CTM with significant bias. **Figure 6** shows the number of days in a year exceeding 100 μg/m³ averaged over 2014–2018 for both the downscaled



and unadjusted EMEP4UK datasets, with yellow cells highlighting areas where 100 μg/m³ is exceeded less than 10 times per year and therefore passing the government objective. We find that 27% of the downscaled UK surface exceeds the government objective, compared to 99% from EMEP4UK. This underpins the importance of bias correction when using process models to examine policy metrics and air quality exposure indicators. At least one downscaled cell in all the 12 UK regions was found to have more than 10 days with MDA8 greater than 100 μg/m³ averaged over 2014–2018, however the regions in the southeast of the UK have the greatest proportion of failing cells, with 86% and 88% of the East and South East regions failing.

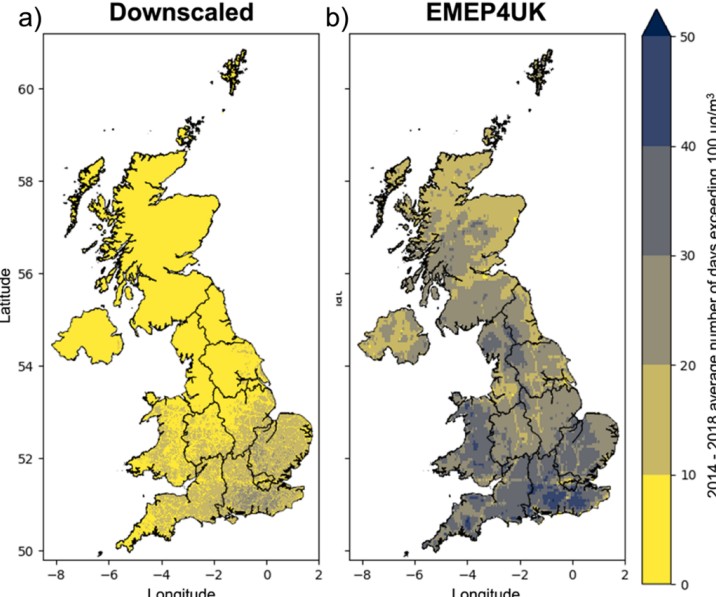

**Figure 6: 2014–2018 average number of days per year where each cell exceeds a level of 100 μg/m³ in a) downscaled ozone surface and b) EMEP4UK ozone surface.**

**4.2 Trends over time**

As mentioned earlier, the use of measurement data only is limited by varying, and in some cases very short, measurement periods and this can prove problematic in air quality trend analysis (Lang, 2020). The gridded downscaled and EMEP4UK datasets facilitate the estimation of trends for all regions, regardless of the density of the measurement network and/or the completeness of the measurement records. We illustrate this in the subsequent analysis by quantifying regional trends in ozone concentrations, comparing, as before, measurement, downscaled and EMEP4UK estimates. We do not quantify a single UK-wide trend since the behaviour of ozone, and consequently also the observed long-term trends, differ considerably across the UK.

A benefit of using a gridded ozone surface is greater spatial coverage; this enables estimation of regional trends in regions where measurement stations are sparse. **Figure 7** shows the annual and March–August regional mean MDA8 ozone trends for 2001–2018, for the three datasets. These trends are also given in Table A5 and Table A6, respectively. All regions have a decreasing trend in annual mean MDA8 ozone in the downscaled surface; however, no trends are statistically significant. Southeast England has the greatest trend at -0.26 [-0.56, 0.04], followed by Southwest England and Wales, at -0.25 [-0.55, 0.04] and -0.25 [-0.50, 0.00] μg/m³, respectively. In comparison, the measurements and EMEP4UK surfaces both have a combination of increasing and decreasing regional trends (few of which are statistically significant). In the measurements, Yorkshire and The Humber has the greatest increasing trend in annual mean ozone concentrations, at 0.33 [0.02, 0.64] μg/m³ per year, followed by the West Midlands and Northwest England at 0.28 [-0.01, 0.56] and 0.29 [-0.02, 0.59] μg/m³ per year, respectively. EMEP4UK is the only dataset for which London has a significant increasing trend in annual mean ozone, at 0.43 [0.20, 0.66] μg/m³, with non-significant decreasing trends of -0.20 [-0.48, 0.09] and -0.23 [-0.50, 0.05] μg/m³ per year in, respectively, the measurements and downscaled surface.



In contrast to the annual mean case discussed above, most March–August mean ozone trends are statistically significant in the downscaled and EMEP4UK surfaces (Table A6). When comparing the March–August mean trends we also see a greater similarity between the downscaled and EMEP4UK surfaces (in terms of the sign of the trend), except for London where the trend is again positive in the EMEP4UK surface (0.17 [-0.06, 0.39] µg/m³ per year), and negative in the downscaled surface (-0.34 [-0.77, 0.10] µg/m³ per year). While all regions have a decreasing March–August mean trend in the downscaled surface, the largest reductions are seen in the south of the UK: Southeast England and Southwest England at -0.58 [-1.02, -0.15] and -0.52 [-0.95, -0.09] µg/m³ per year, respectively. Note, the MDA8 ozone data that underpin the above trend analysis is shown in Figures A3–A5 and A6–A8 for the annual mean and March–August mean, respectively. These figures demonstrate the regional interannual variation in MDA8 ozone concentrations, which is missed by only considering the trends.

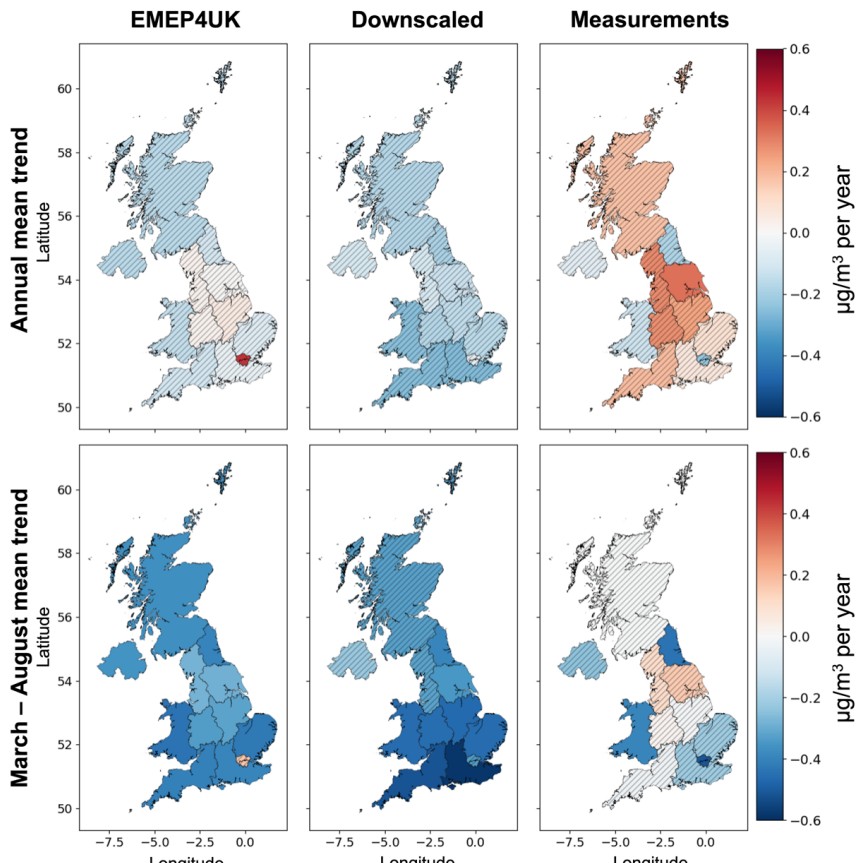

**Figure 7: Annual mean and March–August mean trends for each region, for each dataset. Regions with insignificant trends are hatched.**

**Figure 8** shows the regional trends of 90th and 10th percentile ozone concentrations for the period 2001–2018; estimates and confidence intervals are also given in Table A7 and Table A8. 90th percentile ozone is decreasing for all regions and all datasets. Looking at the downscaled surface, the downward trend is significant in half of the regions considered, with the greatest changes in 90th percentile ozone in the south of the UK, particularly for South East (England) with a trend of -0.74 [-1.35, -0.12] µg/m³ per year. In comparison, regions with the greatest change in 90th percentile ozone in the EMEP4UK surface and measurements are East of England at -0.75 [-1.09, -0.40], and North East (England) at -0.59 [-0.95, -0.22] µg/m³ per year. 10th percentile ozone is increasing for most regions in the downscaled and EMEP4UK surface, and for all regions in the measurements. Northern Ireland, Wales, and Scotland have a slightly decreasing 10th percentile trends in the downscaled surface, at -0.01, -0.01 and 0.06 µg/m³ per year, respectively, though none of these trends are statistically significant. We find a greater increase in 10th percentile ozone for London in the downscaled surface than in the measurements, at 1.19 [0.75,








1.62] and 0.17 [-0.04, 0.37] µg/m³ per year, respectively. We suspect this is again due to urban site type bias in the measurements, compared with the more varied sampling in the gridded downscaled surface. The regional yearly 90th percentiles in the downscaled, EMEP4UK and measurement datasets are shown in Figure A9, Figure A10 and Figure A11, respectively. The equivalent figures for the 10th percentiles are shown in Figure A12, Figure A13 and Figure A14.


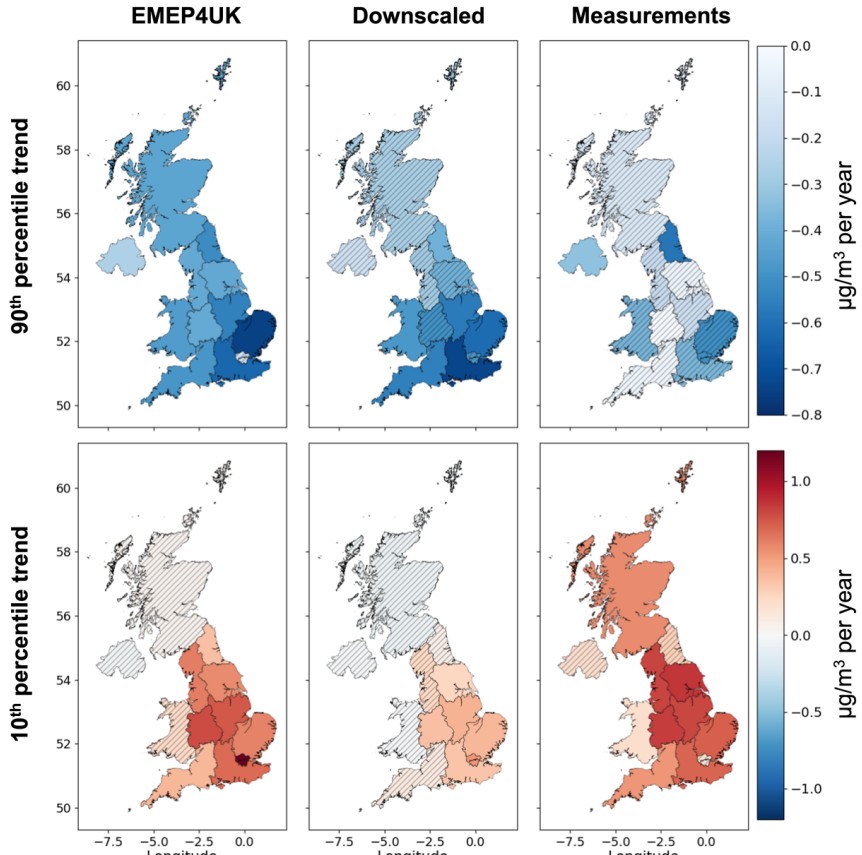

**Figure 8: 90th and 10th percentile trends for each region, for each dataset. Regions with insignificant trends are hatched.**

The UK government has a long-term objective that MDA8 ozone should not exceed a level of 100 µg/m³ more than ten times per year. Earlier we showed that conclusions regarding the degree to which this objective is being met differ substantially between the downscaled and original EMEP4UK model (**Figure 6**). As the downscaled surface, measurement data and EMEP4UK surface all have a different number of cells/stations, we look at the time trend in the percentage of sites or grid cells which fail to meet this, rather than the trend in absolute number. A decreasing time trend in the percentage of the UK failing to meet the objective is seen for all three datasets. However, the only statistically significant trend is for the downscaled surface, at -2.19 [-4.32, -0.07] % per year. The EMEP4UK trend is less steep, -0.60 [-1.62, 0.43] % per year, and the measurements trend lies in between at -1.73 [-3.78, 0.32] % per year.

Recalling that a core aim for our downscaling methodology was to better represent the tail behaviour of measurement ozone, we now consider the specific years 2003, 2006 and 2018 (hereafter "heatwave years") which were significantly warmer than average (see later), and when UK ozone levels were elevated (Diaz et al., 2020). **Figure 9** shows the number of days that exceed a level of 100 µg/m³ for each heatwave year, along with the corresponding yearly mean temperature. Yellow cells highlight areas where 100 µg/m³ is exceeded less than 10 times per year and therefore passing the government objective. In the EMEP4UK surface, almost all the UK (more than 99%) is exceeding the government objective in the heatwave years. In the downscaled surface, 88%, 87%



and 53% of the UK is failing the government objective in 2003, 2006 and 2018, respectively. These percentages are substantially higher than the 2014–2018 average percentage of 27%, demonstrating the more frequent occurrence of exceedance days above 100 μg/m³ in heatwave years. Both the EMEP4UK and downscaled surfaces show a change over time in the amount of the UK exceeding the government objective in heatwave years, with the highest number of exceedances in 2003, and lowest in 2018. The areas with the highest number of exceedances are correlated with the temperature maps, with more exceedance occurring where the yearly mean temperatures are higher, consistent with the well documented link between MDA8 ozone and temperature in the literature.

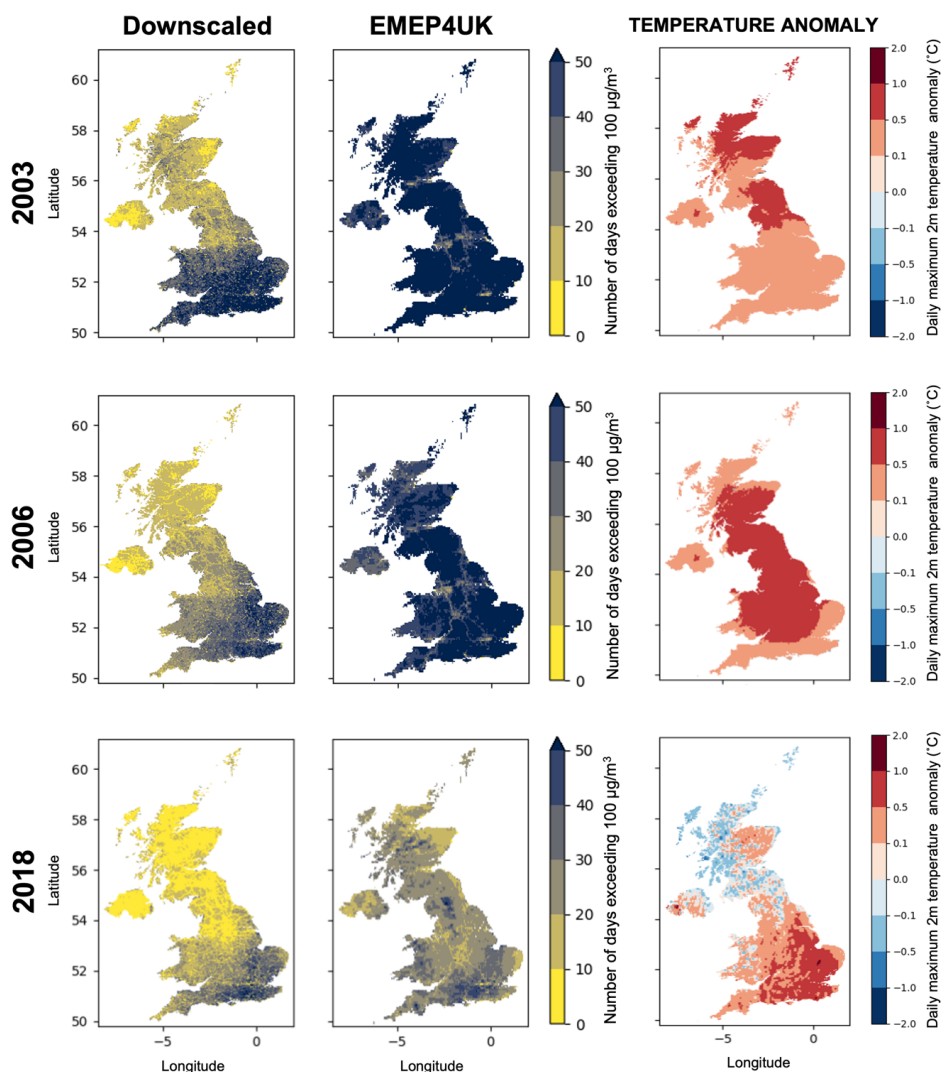

**Figure 9: Number of days where each cell exceeds a level of 100 μg/m³ in downscaled surface ozone (left) and EMEP4UK surface ozone (middle), for heatwaves years 2003, 2006 and 2018. Also given are the yearly mean daily maximum temperature anomalies compared to the 2001–2018 average (right) for each heatwave year.**



### 4.3 Analysis of NO$_X$ scenarios

A major application of CTMs is to aid understanding of pollutant behaviour under future emissions or climate change scenarios. In this section we explore the effect of reductions in UK NO$_X$ emissions on ozone under 2018 meteorological conditions. We compare four downscaled surfaces: (1) a 2018 base run, (2) a run as base but with a 20% reduction in UK terrestrial NO$_X$ emissions, (3) a 40% reduction, and (4) an 80% reduction (see also Section 2.1). The year 2018 is of specific interest as it was the seventh warmest year in the UK since 1884 (Kendon et al., 2019) with a mean temperature that was 0.6 °C above the 1981–2010 average.

Since 2018 was, climatologically, an atypical year for the UK, we first compared two 2018 EMEP4UK base run downscaled surfaces. The first was obtained from the ML downscaling model trained on the 2001–2018 EMEP4UK data used in Sections 3 and 4, and the second from the same downscaling model but trained only on the 2018 base run data. We found that the two sets of predicted surfaces performed almost identically in capturing the behaviour of the 2018 measurement data (both have an R$^2$ of 0.74 and RMSE of ~12.15 μg/m$^3$). These findings support our decision to use the downscaling model trained on the 2001–2018 data (see Section 3) to downscale not just the base run of 2018, but also the ozone surfaces under the three NO$_X$ scenario runs. Implicit in downscaling the scenarios in this way is the assumption that the associations of the input features and surface level ozone remain the same, even as NO$_X$ levels decrease. We acknowledge this as a potential limitation but note that similar assumptions are endemic throughout the downscaling literature.

**Figure 10** shows the point wise differences in downscaled annual mean, March–August mean, and 90[th] percentile ozone for the three NO$_X$ scenarios compared to the equivalent statistics for the 2018 base run. Both 20% and 40% NO$_X$ reductions result in most of the UK seeing increased annual mean ozone concentrations, except rural areas in Scotland. Conversely, the more drastic NO$_X$ reduction of 80% results in a decrease of annual mean ozone for much of the UK, particularly large portions of South West (England), Wales, Scotland, Northern Ireland, North West (England) and North East (England). In contrast to the trends seen in the annual mean, larger and more widely spread decreases are seen in the March–August mean for all NO$_X$ reduction scenarios, suggesting the impact on spring and summer mean ozone is greater than on the annual mean. The change in 90[th] percentile ozone is far more granular, largely due to the differences in tail behaviour of ozone at rural and urban locations. The relatively moderate reductions in NO$_X$ concentrations of 20% and 40% lead to increases in 90[th] percentile ozone in parts of the UK, whereas the more substantial reduction in NO$_X$ concentrations of 80% results in only very urban areas having increases in 90[th] percentile ozone, such as Manchester, Leeds, Sheffield, Birmingham, London, Newcastle, Edinburgh, Glasgow, and Aberdeen. Similar increases in high-level ozone due to NO$_X$ reductions has been show for several cities in the US (Gao et al., 2013) and reflect the interdependence of ozone concentrations and NO$_X$ mitigation strategies. Finally, a similar sensitivity analysis based on the unadjusted EMEP4UK output (Figure A15) exhibits a very similar pattern of ozone response.

In summary, the above analysis demonstrates the applicability of our downscaled EMEP4UK surface to examine the sensitivity of UK ozone to changes in precursor emissions at high spatial resolution. These results also emphasise the challenges in controlling surface ozone (especially in urban areas) if NOx emissions continue to decline substantially – an effect which few studies have demonstrated for the UK using models to date.



545

**Figure 10: Difference in annual mean (top), March–August mean (middle), and 90th percentile (bottom) MDA8 ozone compared to 2018 for three UK NOX scenarios: 20% reduction in NOX (left), 40% reduction in NOX (middle), 80% reduction in NOX (right).**



## 5 Conclusions

We have proposed a machine learning methodology to spatially downscale surface ozone output from the EMEP4UK chemical transport model from its native 5×5 km resolution to a 1×1 km resolution. Taking a 1×1 km interpolation of the original EMEP4UK grid as input, our algorithm uses a gradient boosting tree to predict a high-resolution gridded ozone surface. The algorithm was trained to predict measurement data from sites across the UK, and in addition to the EMEP4UK data has 21 input feature variables. We find that the downscaled surface better represents the behaviour of measurement ozone, with a 128% improvement in $R^2$ and 37% reduction in RMSE compared to the EMEP4UK surface. The GBT allows replication of the behaviour of complex non-linear systems and the ability to work with high dimensional datasets. Producing the downscaled surface using the proposed methodology is far quicker and less computationally expensive than running a high-resolution CTM. We therefore consider this methodology to be a useful post-processing tool for CTMs that can efficiently produce higher resolution ozone surfaces and, as is the case for EMEP4UK, reduce biases by incorporating information from measurements. A further advantage of the proposed ML downscaling model is the ability to identify the most important features for the prediction of MDA8 ozone. Consistent with previous work, daily maximum 2m temperature is found to be the most important meteorological feature, with elevated temperatures strongly associated with high level ozone.

Our analysis on recent years (2014–2018) finds that South East (England) and South West (England) experience higher March–August concentrations of ozone than other regions. We find greater inter-region differences in spring/summer mean ozone concentrations than annual mean. There is a clear north-south difference of high percentile ozone in the downscaled surface, with high ozone concentrations in the south of the UK. Low percentile ozone has the greatest inter-region variation in the downscaled surface due to the particularly low $10^{th}$ percentile ozone concentration in London. This demonstrates the effect greater $NO_X$ concentrations in highly urban areas in reducing background ozone concentrations through $NO_X$-titration.

We have estimated regional trends in various statistics of ozone using data from 2001–2018 for the three datasets: EMEP4UK, downscaled surface, and measurements. Annual and March–August mean ozone decreases for all regions in the downscaled surface, while some regions have increasing trends in the measurements. EMEP4UK is the only dataset to estimate an increase in annual mean ozone for London. The proposed downscaling surface is useful when considering how UK ozone has changed over time as it is higher resolution than EMEP4UK, and provides more spatially complete coverage than measurements alone. The downscaling process also addresses the high bias present in EMEP4UK, resulting in a better reflection of high-level ozone relevant to health. We find an improved picture of high-level ozone when using the downscaled surface, with only 53% of the UK failing its government objective (to not exceed an ozone level of 100 μg/m³ more than 10 times per year) in 2018, compared to 99% of the UK failing this objective in EMEP4UK. Further improvement in high-level ozone is apparent from considering trends in $90^{th}$ percentile ozone. We find significant reductions in $90^{th}$ percentile ozone for half of the regions considered in the downscaled surface, with the greatest reductions in the south of the UK, particularly for South East (England).

Through a sensitivity analysis, we considered the effect of three $NO_X$ reduction scenarios on UK ozone concentrations downscaled using the proposed downscaling method. Moderate (20% and 40%) reductions in $NO_X$ concentrations are shown to *increase* annual mean ozone for most of the UK, whereas significant (80%) reductions *decrease* annual mean ozone for large parts of the UK. More of the UK shows a decrease in March–August mean ozone for all $NO_X$ scenarios, suggesting a stronger link between spring and summer ozone concentrations and $NO_X$ than annual mean concentrations. The differences in the tail behaviour of ozone at urban and rural locations is made evident in the effect of $NO_X$ reductions on $90^{th}$ percentile ozone. Very urban areas see the largest increases in $90^{th}$ percentile ozone when reducing $NO_X$ concentrations by 80%, this includes many of the UK's biggest cities. We determine it important to further understand the effect of $NO_X$ reductions on UK ozone, as a considerable portion of the UK population live in these urban areas. These results reemphasise the broader challenges around $NO_X$ mitigation strategies. To conclude, machine learning based downscaling approaches offer a promising way to study pollutant trends and to assess the impact of policies for ozone and in principle other pollutants. A focus of future work will be to exploit the bias-corrected downscaled surfaces for the assessment of population exposure to poor air quality and to help quantify the resulting health impacts.

## Code and data availability

The EMEP4UK ozone data is available for 2001–2015 from https://catalogue.ceh.ac.uk/documents/b0545f67-e47c-4077-bf3c-c5ffcd6b72c8; 2016–2018 ozone data and $NO_X$ scenario data is available upon request. The XGBoost code used is available from https://doi.org/10.1145/2939672.2939785 (Chen and Guestrin, 2016). The measurement data is available through the R package *openair*: https://davidcarslaw.github.io/openair/.



**Author contribution**

**Lily Gouldsbrough**: Conceptualization, Methodology, Data curation, Software, Formal analysis, Writing – original draft, Visualization, Investigation. **Emma Eastoe**: Conceptualization, Methodology, Writing – review & editing, Supervision. **Ryan Hossaini**: Conceptualization, Methodology, Writing – review & editing, Supervision. **Paul J. Young**: Conceptualization, Writing – review & editing, Supervision. **Massimo Vieno**: Data curation, Writing – review and editing.

**Acknowledgements**

LG acknowledges the UK Engineering and Physical Research Council (EPSRC) for a PhD studentship (EP/R513076/1, project reference 2353903). RH was supported by a UK Natural Environment Research Council (NERC) Independent Research Fellowship (NE/N014375/1). EE and PJY were supported by the EPSRC-funded Data Science of the Natural Environment project (EP/R01860X/1). MV was supported by the Natural 620 Environment Research Council (NE/R016429/1) as part of the UK-SCAPE programme delivering National Capability.

**Competing interests**

The authors declare that they have no conflict of interest.

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

**Appendix**

This appendix contains additional figures and data to supplement the information in the main text.

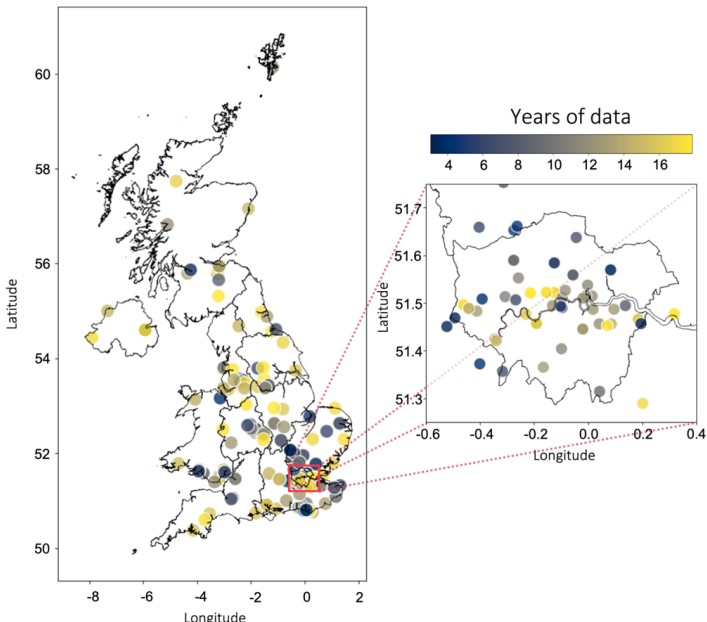

**Figure A1: Measurement station map with number of years of data used for each station.**



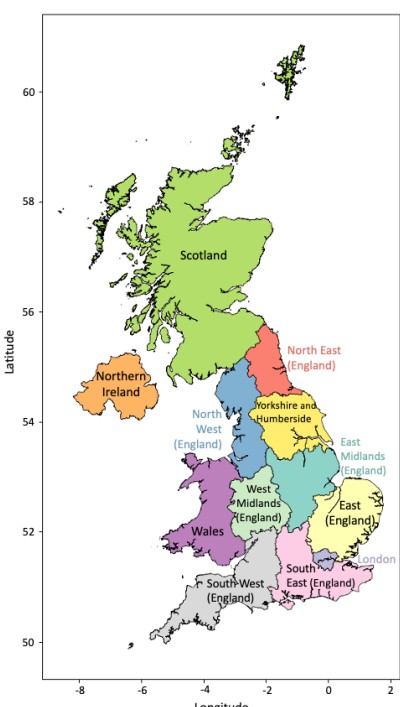

**Figure A2: The region definitions for this paper (Level 1 Nomenclature of Territorial Units for Statistics).**


**Table A1: Annual mean MDA8 (µg/m³) per region for 2001–2018, with 95% confidence intervals of the mean estimate shown in square brackets.**

| Region | Downscaled | EMEP4UK | Measurements |
|---|---|---|---|
| East Midlands (England) | 62.38 [62.37, 62.38] | 75.02 [74.99, 75.05] | 62.15 [61.76, 62.54] |
| East of England | 64.96 [64.95, 64.96] | 75.89 [75.86, 75.93] | 64.55 [64.23, 64.87] |
| London | 57.37 [57.34, 57.39] | 70.41 [70.31, 70.54] | 48.56 [48.37, 48.75] |
| North East (England) | 60.68 [60.68, 60.69] | 77.31 [77.27, 77.35] | 58.79 [58.30, 59.24] |
| North West (England) | 59.72 [59.72, 59.73] | 76.39 [76.35, 76.42] | 58.76 [58.46, 59.10] |
| Northern Ireland | 59.16 [59.15, 59.16] | 77.44 [77.42, 77.46] | 59.40 [58.97, 59.85] |
| Scotland | 62.77 [62.77, 62.77] | 78.30 [78.29, 78.31] | 66.14 [65.92, 66.37] |
| South East (England) | 66.24 [66.24, 66.25] | 76.82 [76.79, 76.84] | 66.49 [66.28, 66.73] |
| South West (England) | 67.11 [67.11, 67.12] | 78.71 [78.70, 78.74] | 65.43 [65.05, 65.81] |
| Wales | 66.61 [66.61, 66.62] | 79.29 [79.26, 79.31] | 64.66 [64.36, 64.92] |
| West Midlands (England) | 62.67 [62.66, 62.67] | 75.00 [74.95, 75.03] | 60.90 [60.58, 61.22] |
| Yorkshire and The Humber | 59.27 [59.26, 59.27] | 74.90 [74.88, 74.93] | 58.97 [58.57, 59.35] |
| **All-region mean** | **62.41 [62.40, 62.42]** | **76.29 [76.26, 76.33]** | **61.23 [60.91, 61.56]** |
| **All-region sdev. (% of mean)** | **3.13 (5.01)** | **2.37 (3.10)** | **4.99 (8.15)** |



**Table A2: March–August mean MDA8 ozone (μg/m³) per region for 2001–2018, with 95% confidence intervals of the mean estimate shown in square brackets.**

| Region | Downscaled | EMEP4UK | Measurements |
|---|---|---|---|
| East Midlands (England) | 71.12 [71.11, 71.13] | 86.42 [86.38, 86.44] | 71.35 [70.85, 71.82] |
| East of England | 75.14 [75.13, 75.14] | 88.07 [88.04, 88.10] | 74.17 [73.78, 74.59] |
| London | 68.90 [68.87, 68.93] | 84.37 [84.27, 84.49] | 58.04 [57.78, 58.34] |
| North East (England) | 66.09 [66.08, 66.10] | 85.18 [85.13, 85.22] | 66.16 [65.59, 66.74] |
| North West (England) | 65.57 [65.56, 65.58] | 85.20 [85.17, 85.23] | 66.48 [66.12, 66.90] |
| Northern Ireland | 62.76 [62.75, 62.77] | 82.61 [82.57, 82.65] | 63.35 [62.78, 63.94] |
| Scotland | 66.30 [66.29, 66.30] | 83.66 [83.65, 83.68] | 70.64 [70.34, 70.95] |
| South East (England) | 75.56 [75.55, 75.57] | 87.86 [87.83, 87.89] | 75.07 [74.79, 75.34] |
| South West (England) | 73.05 [73.05, 73.06] | 86.12 [86.10, 86.15] | 71.91 [71.38, 72.46] |
| Wales | 71.51 [71.50, 71.51] | 85.84 [85.81, 85.87] | 69.21 [68.81, 69.58] |
| West Midlands (England) | 70.40 [70.39, 70.40] | 85.06 [85.03, 85.09] | 69.32 [68.89, 69.71] |
| Yorkshire and The Humber | 66.69 [66.69, 66.70] | 85.19 [85.16, 85.22] | 67.02 [66.45, 67.52] |
| **All-region mean** | **69.42 [69.41, 69.43]** | **85.47 [85.43, 85.50]** | **68.56 [68.13, 68.99]** |
| **All-region sdev. (% of mean)** | **4.04 (5.82)** | **1.57 (1.83)** | **4.75 (6.93)** |


**Table A3: Regional average 90th percentiles of MDA8 ozone (μg/m³) for 2001–2018, with 95% confidence intervals of the mean estimate shown in square brackets.**

| Region | Downscaled | EMEP4UK | Measurements |
|---|---|---|---|
| East Midlands (England) | 82.58 [82.59, 82.56] | 97.22 [97.17, 97.26] | 84.87 [84.27, 85.57] |
| East of England | 86.52 [86.53, 86.51] | 98.11 [98.08, 98.14] | 87.32 [86.78, 87.91] |
| London | 82.52 [82.57, 82.48] | 96.71 [96.44, 96.94] | 77.19 [76.88, 77.47] |
| North East (England) | 77.84 [77.85, 77.83] | 96.18 [96.13, 96.24] | 80.22 [79.38, 80.74] |
| North West (England) | 78.32 [78.34, 78.32] | 96.90 [96.86, 96.95] | 81.34 [81.01, 81.76] |
| Northern Ireland | 75.61 [75.62, 75.60] | 95.59 [95.55, 95.63] | 79.00 [78.28, 79.56] |
| Scotland | 79.03 [79.04, 79.03] | 95.41 [95.40, 95.42] | 86.20 [85.90, 86.50] |
| South East (England) | 87.86 [87.87, 87.84] | 99.48 [99.43, 99.53] | 89.48 [89.13, 89.77] |
| South West (England) | 85.53 [85.54, 85.52] | 98.99 [98.95, 99.03] | 89.63 [88.94, 90.07] |
| Wales | 83.88 [83.89, 83.87] | 98.72 [98.69, 98.75] | 86.41 [86.00, 86.84] |
| West Midlands (England) | 81.90 [81.91, 81.89] | 97.22 [97.18, 97.28] | 84.07 [83.59, 84.78] |
| Yorkshire and The Humber | 78.38 [78.39, 78.37] | 96.23 [96.19, 96.28] | 82.52 [81.60, 83.26] |
| **All-region mean** | **81.66 [81.65, 81.68]** | **97.23 [97.17, 97.29]** | **84.02 [83.48, 84.52]** |
| **All-region sdev (% of mean)** | **3.86 (4.72)** | **1.34 (1.37)** | **4.04 (4.80)** |


**Table A4: Regional average 10th percentiles of MDA8 ozone (μg/m³) for 2001–2018, with 95% confidence intervals of the mean estimate shown in square brackets.**

| Region | Downscaled | EMEP4UK | Measurements |
|---|---|---|---|
| East Midlands (England) | 41.52 [41.50, 41.53] | 52.02 [51.94, 52.09] | 37.09 [36.38, 37.79] |
| East of England | 42.05 [42.04, 42.07] | 51.79 [51.71, 51.87] | 39.25 [38.69, 40.04] |
| London | 29.18 [29.12, 29.23] | 40.57 [40.25, 40.79] | 16.96 [16.65, 17.27] |
| North East (England) | 45.32 [45.31, 45.33] | 58.97 [58.88, 59.04] | 36.11 [34.80, 36.80] |
| North West (England) | 41.71 [41.70, 41.73] | 56.10 [56.01, 56.17] | 31.52 [30.72, 32.29] |
| Northern Ireland | 44.42 [44.41, 44.43] | 60.97 [60.92, 61.00] | 38.88 [38.27, 39.61] |
| Scotland | 48.31 [48.31, 48.31] | 61.49 [61.47, 61.51] | 45.34 [44.99, 45.66] |
| South East (England) | 44.13 [44.12, 44.15] | 54.18 [54.10, 54.24] | 41.09 [40.63, 41.55] |
| South West (England) | 49.22 [49.21, 49.22] | 60.95 [60.91, 60.99] | 38.72 [37.78, 39.48] |
| Wales | 50.28 [50.27, 50.29] | 62.08 [62.04, 62.11] | 39.90 [39.17, 40.42] |
| West Midlands (England) | 42.27 [42.26, 42.29] | 53.48 [53.40, 53.55] | 35.86 [35.37, 36.63] |
| Yorkshire and The Humber | 40.31 [40.30, 40.33] | 52.82 [52.74, 52.89] | 34.63 [33.80, 35.45] |
| **All-region mean** | **43.23 [43.21, 43.24]** | **55.45 [55.36, 55.52]** | **36.28 [35.60, 36.92]** |
| **All-region sdev (% of mean)** | **5.49 (12.70)** | **6.13 (11.05)** | **7.00 (19.30)** |



**Table A5: Annual mean trend of MDA8 ozone (µg/m³/yr) per region for 2001–2018, with 95% confidence intervals of the mean estimate shown in brackets. Significant trends are in bold.**

| Region | Downscaled | EMEP4UK | Measurements |
|---|---|---|---|
| East Midlands (England) | -0.15 [-0.42, 0.13] | 0.05 [-0.16, 0.26] | 0.24 [-0.08, 0.55] |
| East of England | -0.16 [-0.46, 0.15] | -0.07 [-0.27, 0.12] | 0.09 [-0.19, 0.36] |
| London | -0.06 [-0.38, 0.26] | **0.43 [0.20, 0.66]** | -0.23 [-0.50, 0.05] |
| North East (England) | -0.19 [-0.41, 0.03] | -0.10 [-0.30, 0.09] | -0.18 [-0.42, 0.06] |
| North West (England) | -0.09 [-0.34, 0.15] | 0.03 [-0.18, 0.24] | 0.29 [-0.02, 0.59] |
| Northern Ireland | -0.10 [-0.32, 0.12] | -0.16 [-0.36, 0.04] | -0.06 [-0.27, 0.15] |
| Scotland | -0.17 [-0.40, 0.05] | -0.16 [-0.38, 0.06] | 0.18 [-0.04, 0.39] |
| South East (England) | -0.26 [-0.56, 0.04] | -0.04 [-0.26, 0.17] | 0.06 [-0.24, 0.37] |
| South West (England) | -0.25 [-0.55, 0.04] | -0.11 [-0.32, 0.10] | 0.19 [-0.10, 0.47] |
| Wales | -0.25 [-0.50, 0.00] | -0.14 [-0.34, 0.05] | -0.13 [-0.36, 0.11] |
| West Midlands (England) | -0.17 [-0.45, 0.11] | 0.03 [-0.18, 0.23] | 0.28 [-0.01, 0.56] |
| Yorkshire and The Humber | -0.13 [-0.35, 0.10] | 0.01 [-0.17, 0.19] | **0.33 [0.02, 0.64]** |


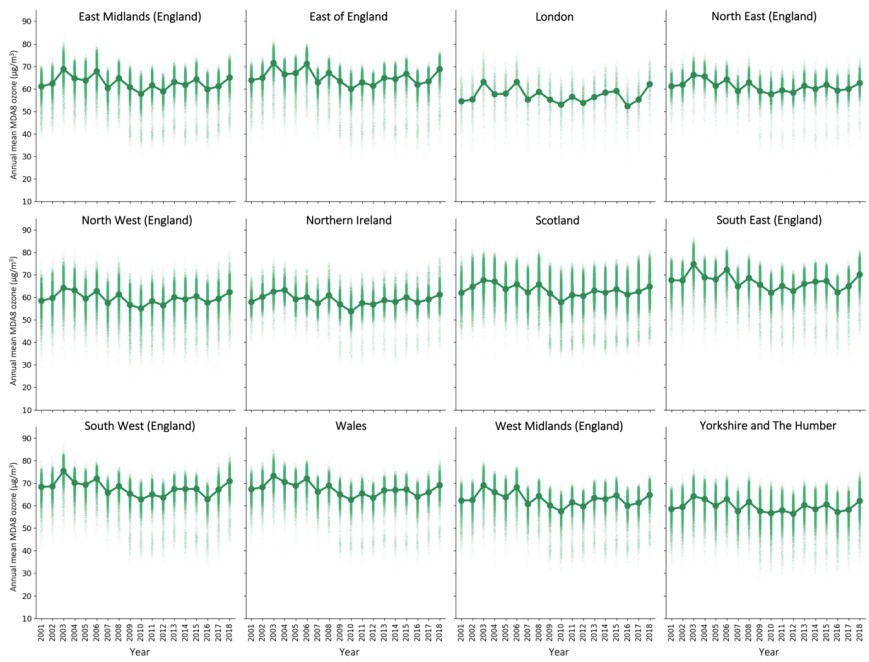

**Figure A3: Regional annual mean MDA8 ozone for the downscaled surface, for 2001–2018. Background dots are individual cell estimates, larger foreground dots are the yearly average.**



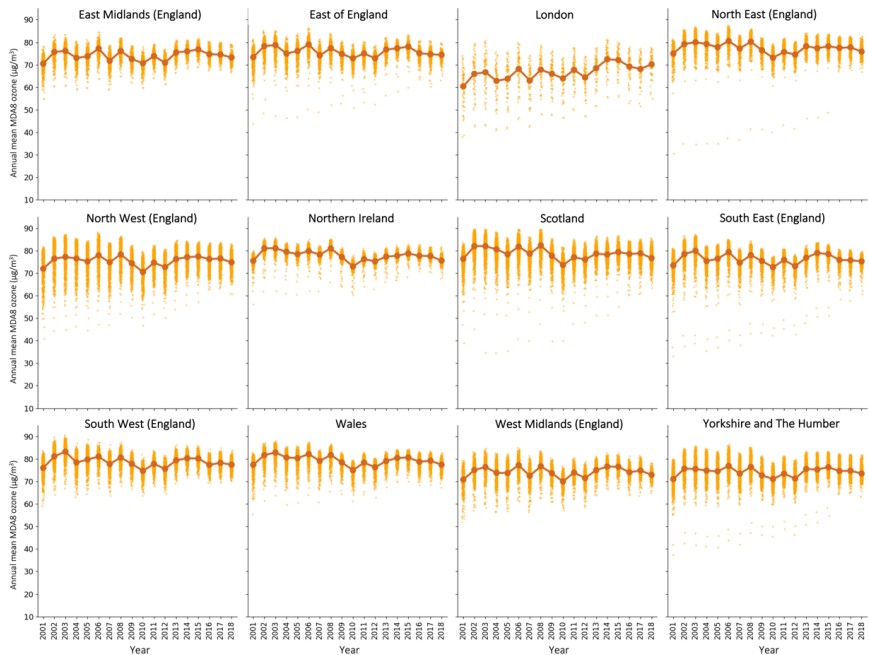

**Figure A4: Regional annual mean MDA8 ozone for the EMEP4UK surface, for 2001–2018. Background dots are individual cell estimates, larger foreground dots are the yearly average.**




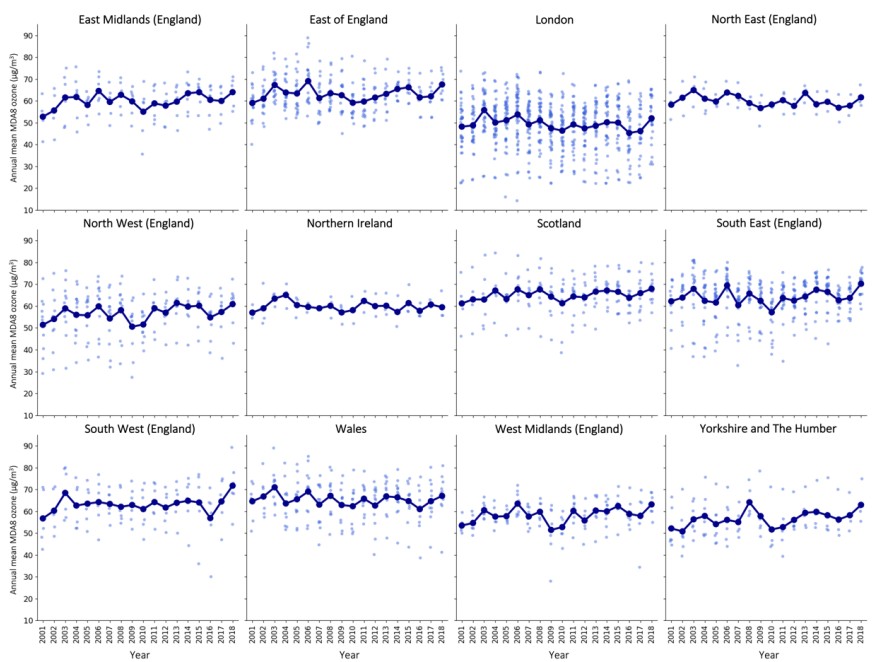

**Figure A5: Regional annual mean MDA8 ozone for the measurement data, for 2001–2018. Background dots are individual cell estimates, larger foreground dots are the yearly average.**

Table A6: March–August mean trends of MDA8 ozone (µg/m³/yr) per region for 2001–2018, with 95% confidence intervals of the mean estimate shown in brackets. Significant trends are in bold.

| Region | Downscaled | EMEP4UK | Measurements |
|---|---|---|---|
| East Midlands (England) | **-0.47 [-0.86, -0.07]** | **-0.32 [-0.56, -0.08]** | -0.01 [-0.42, 0.40] |
| East of England | **-0.46 [-0.88, -0.04]** | **-0.43 [-0.65, -0.21]** | -0.21 [-0.60, 0.18] |
| London | -0.34 [-0.77, 0.10] | 0.17 [-0.06, 0.39] | **-0.51 [-0.89, -0.13]** |
| North East (England) | **-0.40 [-0.71, -0.08]** | **-0.40 [-0.62, -0.18]** | **-0.44 [-0.76, -0.12]** |
| North West (England) | -0.32 [-0.65, 0.02] | **-0.28 [-0.50, -0.06]** | 0.11 [-0.26, 0.47] |
| Northern Ireland | -0.22 [-0.53, 0.09] | **-0.36 [-0.59, -0.13]** | -0.24 [-0.51, 0.02] |
| Scotland | -0.32 [-0.67, 0.02] | **-0.37 [-0.62, -0.11]** | 0.00 [-0.33, 0.33] |
| South East (England) | **-0.58 [-1.02, -0.15]** | **-0.39 [-0.62, -0.16]** | -0.21 [-0.63, 0.2] |
| South West (England) | **-0.52 [-0.95, -0.09]** | **-0.41 [-0.64, -0.18]** | -0.02 [-0.46, 0.41] |
| Wales | **-0.46 [-0.82, -0.10]** | **-0.44 [-0.65, -0.23]** | **-0.40 [-0.77, -0.02]** |
| West Midlands (England) | **-0.46 [-0.85, -0.06]** | **-0.33 [-0.57, -0.10]** | 0.02 [-0.38, 0.41] |
| Yorkshire and The Humber | **-0.35 [-0.68, -0.02]** | **-0.29 [-0.50, -0.08]** | 0.15 [-0.29, 0.59] |



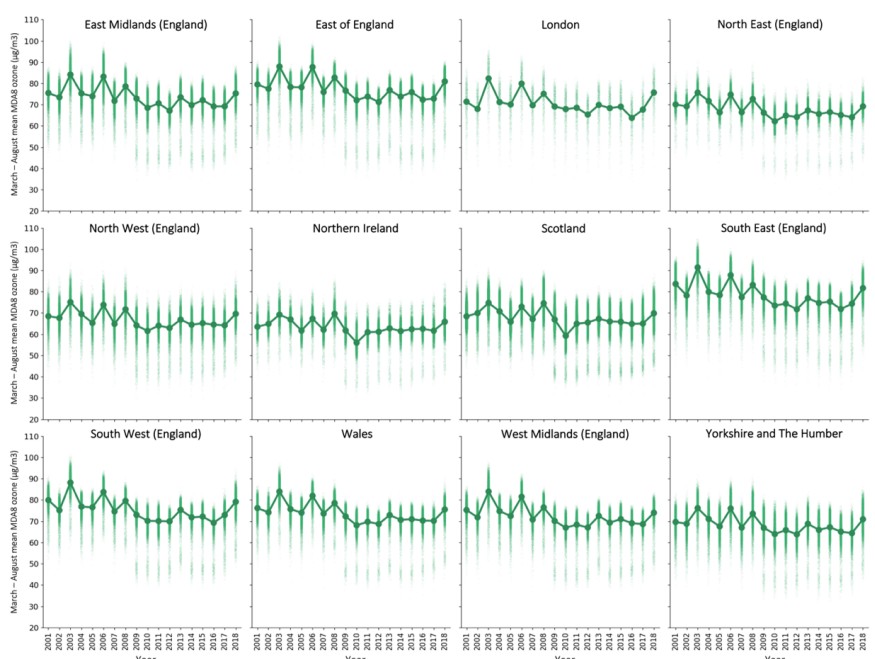

**Figure A6: Regional March–August mean MDA8 ozone for the downscaled surface, for 2001–2018. Background dots are individual cell estimates, larger foreground dots are the yearly average.**

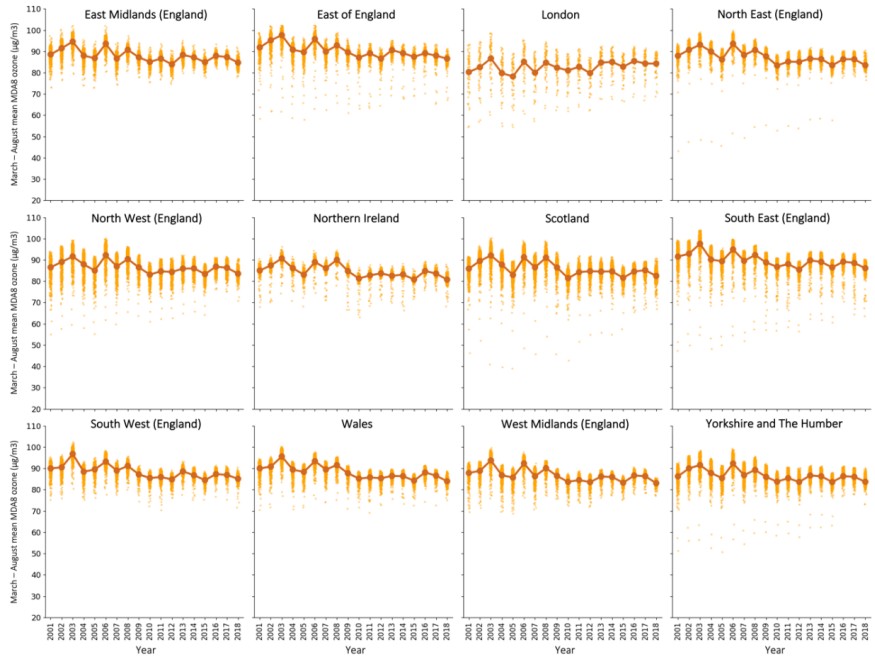

**Figure A7: Regional March–August mean MDA8 ozone for the EMEP4UK surface, for 2001–2018. Background dots are individual cell estimates, larger foreground dots are the yearly average.**



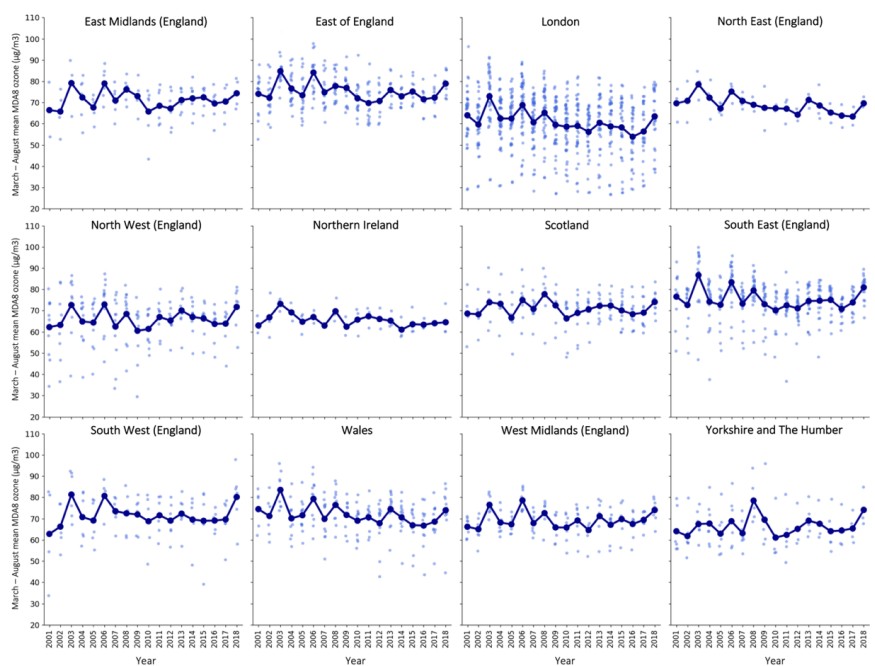

**Figure A8: Regional March–August mean MDA8 ozone for the measurement data, for 2001–2018. Background dots are individual cell estimates, larger foreground dots are the yearly average.**


**Table A7: 90th percentile trends of MDA8 ozone (µg/m³/yr) per region for 2001–2018, with 95% confidence intervals of the mean estimate shown in brackets. Significant trends are in bold.**

| Region | Downscaled | EMEP4UK | Measurements |
|---|---|---|---|
| East Midlands (England) | **-0.57 [-1.12, -0.02]** | **-0.55 [-0.90, -0.20]** | -0.17 [-0.63, 0.28] |
| East of England | **-0.61 [-1.18, -0.04]** | **-0.75 [-1.09, -0.40]** | -0.50 [-1.02, 0.02] |
| London | -0.48 [-1.14, 0.19] | -0.16 [-0.48, 0.16] | **-0.51 [-0.95, -0.07]** |
| North East (England) | **-0.39 [-0.77, -0.01]** | **-0.52 [-0.79, -0.25]** | **-0.59 [-0.95, -0.22]** |
| North West (England) | -0.29 [-0.69, 0.11] | **-0.42 [-0.70, -0.14]** | -0.19 [-0.57, 0.20] |
| Northern Ireland | -0.16 [-0.49, 0.16] | **-0.26 [-0.50, -0.02]** | **-0.33 [-0.62, -0.03]** |
| Scotland | -0.27 [-0.62, 0.09] | **-0.43 [-0.74, -0.13]** | -0.11 [-0.45, 0.22] |
| South East (England) | **-0.74 [-1.35, -0.12]** | **-0.63 [-0.97, -0.28]** | -0.36 [-0.87, 0.14] |
| South West (England) | **-0.55 [-1.06, -0.05]** | **-0.48 [-0.79, -0.18]** | -0.03 [-0.47, 0.40] |
| Wales | **-0.48 [-0.90, -0.06]** | **-0.47 [-0.74, -0.19]** | -0.37 [-0.80, 0.05] |
| West Midlands (England) | -0.50 [-1.02, 0.02] | **-0.41 [-0.74, -0.09]** | 0.01 [-0.42, 0.45] |
| Yorkshire and The Humber | -0.39 [-0.82, 0.05] | **-0.42 [-0.70, -0.15]** | -0.04 [-0.53, 0.46] |



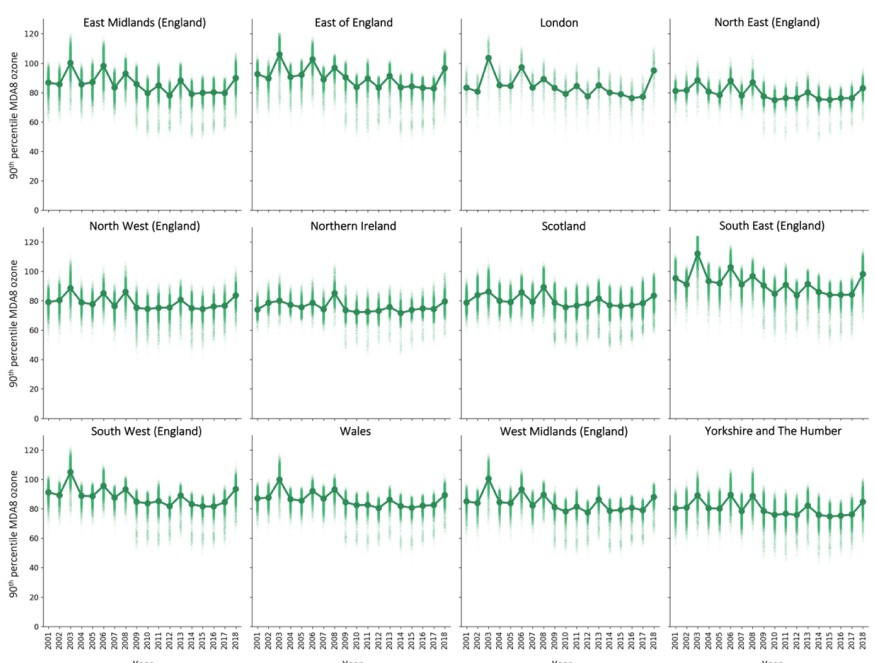

**Figure A9: Regional 90th percentile MDA8 ozone for the downscaled surface, for 2001–2018. Background dots are individual cell estimates, larger foreground dots are the yearly average.**

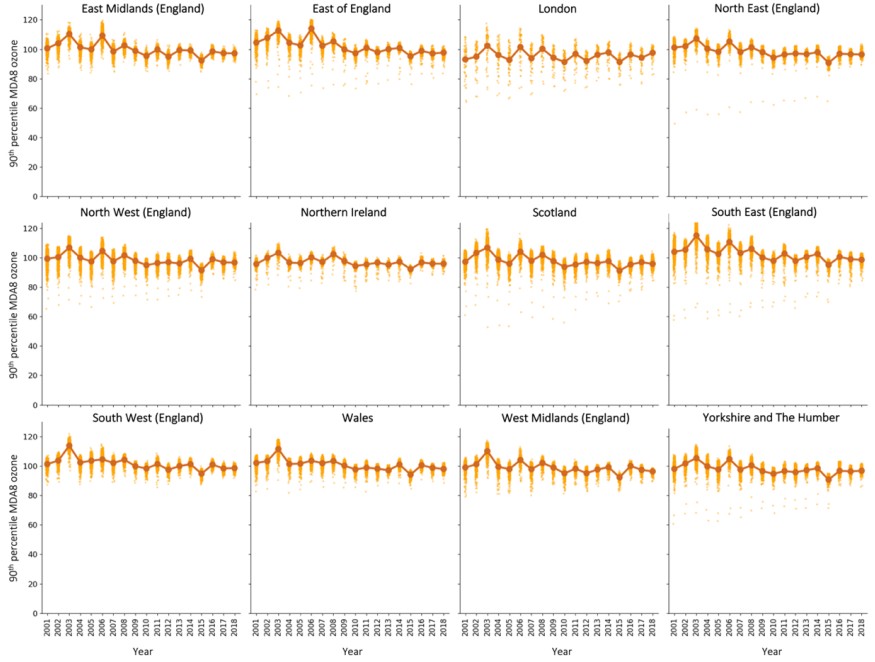

**Figure A10: Regional 90th percentile mean MDA8 ozone for the EMEP4UK surface, for 2001–2018. Background dots are individual cell estimates, larger foreground dots are the yearly average.**



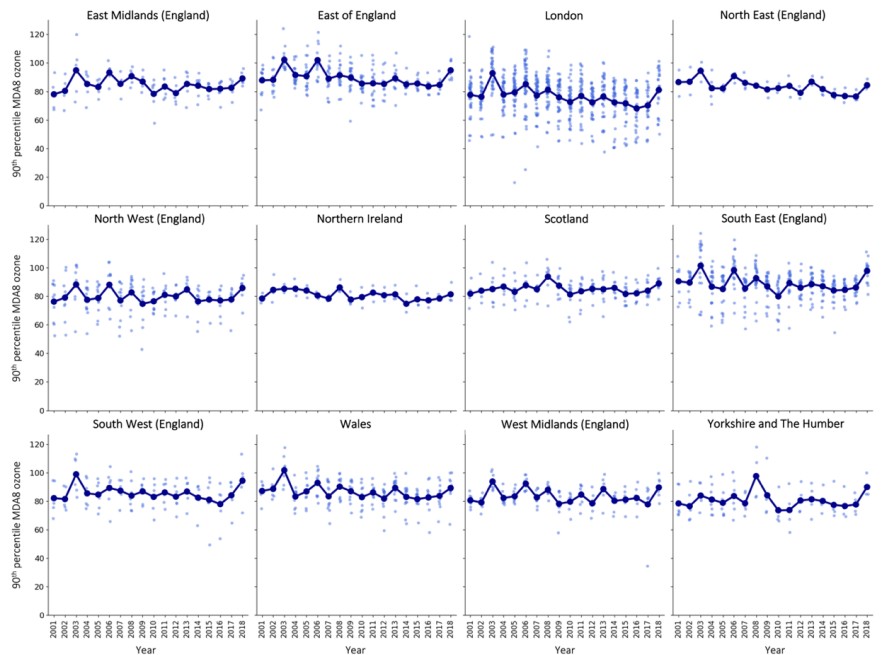


**Figure A11: Regional 90<sup>th</sup> percentile MDA8 ozone for the measurement data, for 2001–2018. Background dots are individual cell estimates, larger foreground dots are the yearly average.**

**Table A8: 10th percentile trends of MDA8 ozone (µg/m³/yr) per region for 2001–2018, with 95% confidence intervals of the mean estimate shown in brackets. Significant trends are in bold.**

| Region | Downscaled | EMEP4UK | Measurements |
|---|---|---|---|
| East Midlands (England) | **0.42 [0.14, 0.69]** | **0.75 [0.40, 1.09]** | **0.80 [0.45, 1.15]** |
| East of England | **0.40 [0.10, 0.69]** | **0.60 [0.27, 0.92]** | **0.71 [0.42, 1.01]** |
| London | **0.53 [0.22, 0.84]** | **1.19 [0.75, 1.62]** | 0.17 [-0.04, 0.37] |
| North East (England) | 0.07 [-0.16, 0.31] | **0.37 [0.05, 0.68]** | 0.28 [-0.03, 0.59] |
| North West (England) | 0.24 [-0.07, 0.54] | **0.61 [0.27, 0.96]** | **0.81 [0.42, 1.20]** |
| Northern Ireland | -0.01 [-0.30, 0.27] | -0.03 [-0.30, 0.24] | 0.20 [-0.12, 0.52] |
| Scotland | -0.06 [-0.29, 0.17] | 0.04 [-0.21, 0.29] | **0.57 [0.27, 0.86]** |
| South East (England) | **0.34 [0.03, 0.64]** | **0.68 [0.30, 1.07]** | **0.70 [0.44, 0.96]** |
| South West (England) | 0.10 [-0.18, 0.38] | **0.40 [0.05, 0.75]** | **0.53 [0.24, 0.82]** |
| Wales | -0.01 [-0.26, 0.24] | 0.23 [-0.08, 0.53] | **0.21 [0.01, 0.41]** |
| West Midlands (England) | **0.36 [0.03, 0.69]** | **0.78 [0.40, 1.16]** | **0.83 [0.48, 1.18]** |
| Yorkshire and The Humber | **0.26 [0.03, 0.49]** | **0.57 [0.27, 0.88]** | **0.86 [0.53, 1.20]** |




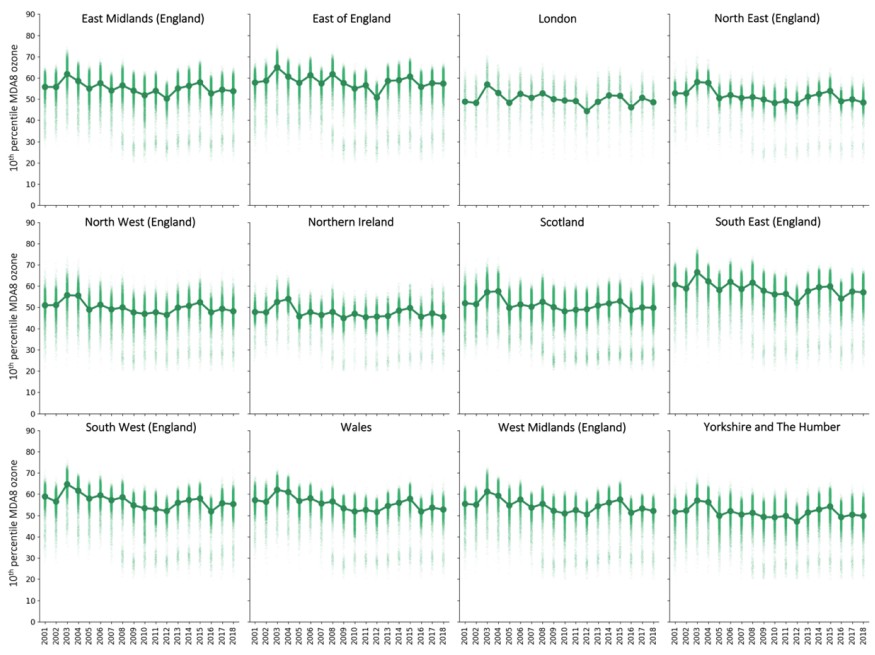

**Figure A12: Regional 10<sup>th</sup> percentile MDA8 ozone for the downscaled surface, for 2001–2018. Background dots are individual cell estimates, larger foreground dots are the yearly average.**

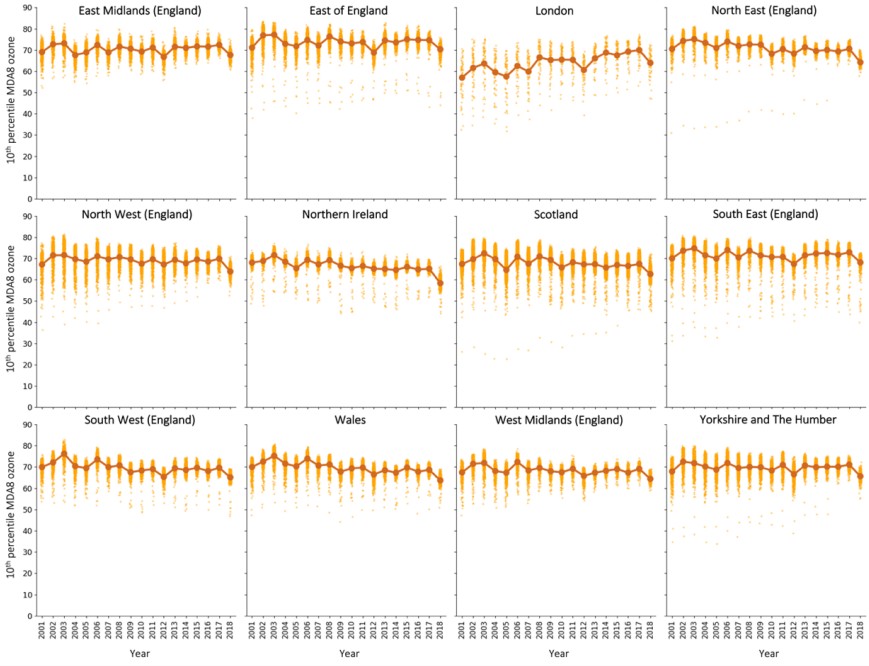

**Figure A13: Regional 10<sup>th</sup> percentile MDA8 ozone for the EMEP4UK surface, for 2001–2018. Background dots are individual cell estimates, larger foreground dots are the yearly average.**



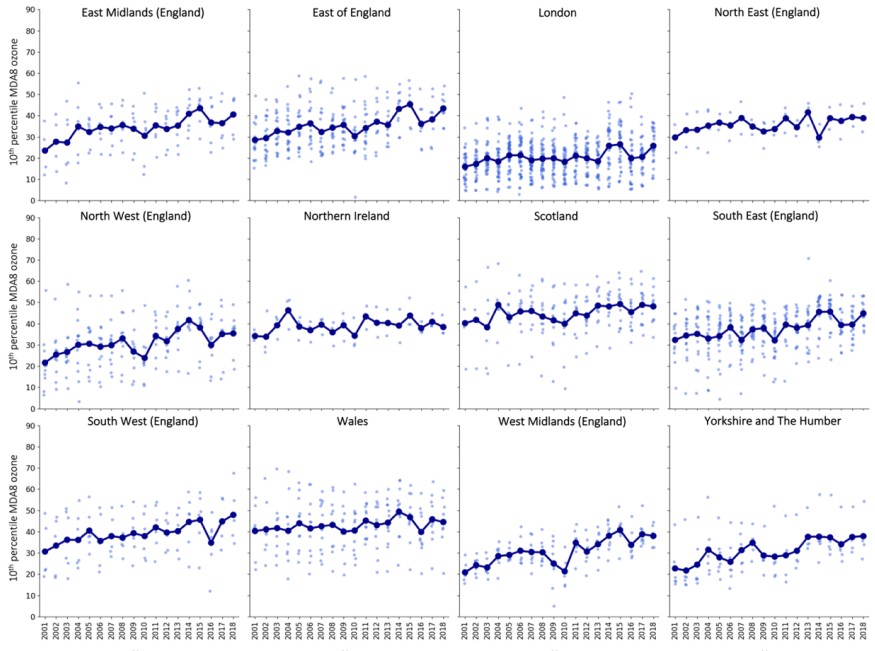

**Figure A14: Regional 10th percentile MDA8 ozone for the measurement data, for 2001–2018. Background dots are individual cell estimates, larger foreground dots are the yearly average.**




**Figure A15:** Original EMEP4UK difference in annual mean (top), March–August mean (middle), and 90th percentile (bottom) MDA8 ozone compared to 2018 for three UK NO$_X$ scenarios: 20% reduction in NO$_X$ (left), 40% reduction in NO$_X$ (middle), 80% reduction in NO$_X$ (right).