# Peer review of "A machine learning approach to downscale EMEP4UK: analysis of UK ozone variability and trends"

_EGUsphere, 2023_

## Author Comment (AC1)

The reviewer's comments are in *blue*, and the author's responses are in *red*. The line numbers quoted in our responses are with reference to the revised manuscript.

**Reviewer 1**

Review of Gouldsbrough "A machine learning approach to downscale EMEP4UK: analysis of UK ozone variability and trends" by Gouldsbrough et al.

The manuscript by Gouldsbrough et al., discusses application of machine learning to downscale ozone results from the EMEP model over the UK. In it, the authors use results from the EMP4UK model along with Gradient Boosting to develop 1x1 km fields of ozone, and their response to emissions. They then use the downscaled model to look at the response of ozone to emissions reductions.

While the paper describes an interesting potential approach to using machine learning (ML) to develop finer scale fields than are typically produced from chemical transport models (CTMs), the current paper has a number of concerns.

I will start out by noting, the paper is pretty well written, and I have few concerns there. Further, I like the idea of using ML to blend observations, CTM results and other data, as they have done here.

**We thank the reviewer for their comments. Responses to their individual comments are given below.**

In terms of limitations, probably the big one is their thought process on how to use the EMP4UK model to look at ozone response after fusing with observations. As noted, the EMEP4UK model is biased, and may be even more biased than indicated in that you should look at bias on a location-by-location basis. The reason bias is SO important is that if you now use a method to remove that bias, you can be artificially shifting in to a different ozone-NOx-VOC response regime. For example, let's say the model is predicting peak ozone levels with a low bias. This would indicate that it is more radical-limited that in might be in actuality. Thus, the response to VOC controls would be enhanced, and NO potentially reduced, if not having the wrong sign. In this application, they have not actually shown that the model response to controls is correct. Without some assurance that the model is correctly capturing the response, using the model as a component of the GB is very concerning.

We thank the reviewer for this interesting observation. However, there is need for clarification. Firstly, let us ignore the bias correction element, and instead consider development of a ML model purely for prediction. In this case, EMEP4UK is not included, and the result is a ML regression tool which can predict ozone fields across the UK using meteorological data and geophysical characteristics of the prediction locations as predictors (feature variables). Such a model must be trained at locations for which there is observational data, but for predictive purposes the model can be extrapolate to any location for which meteorological and geophysical information is available. The limitation of approach is that it would not allow prediction of ozone under different  $NO_X$  scenarios. The next step in complexity is to include measured  $NO_X$  and VOCs as additional feature variables in the MLE

regression model. The resulting model is likely to give more accurate predictions at locations for which  $NO_X$  and VOC measurements are available but is limited for predictive purposes to only those locations for which  $NO_X$  and VOC measurements are available.

To address the limitations of these two approaches, EMEP4UK ozone is included as a feature variable. Whilst we agree that there is unknown uncertainty associated with the EMEP4UK ozone fields, and that this uncertainty may well vary with location, season and ozone regime, both quantification and identification of the sources of this uncertainty is outside the scope of this study and are therefore ignored. We feel justified in doing this as EMEP4UK has previously been validated (Vieno et al., 2010) and is virtually identical to the open source EMEP MSC-W which is currently used to support European policy development by the UNECE Convention on Long-range Transboundary Air Pollution (CLRTAP) and the European Commission [http://www.emep4uk.ceh.ac.uk/].

On this basis, we feel justified in using EMEP4UK ozone as an additional feature variable and allowing the relationship between this and measured ozone to contribute to the prediction of ozone. The GB model uses data to infer relationships between in-situ ozone measurements and 'matched' EMEP4UK model output, these relationships are then used to predict the insitu measurements from EMEP4UK output for locations at which actual in-situ measurements are unavailable. It is true that, in bias correcting the EMEP4UK ozone fields, the resulting ML ozone fields may not be consistent with the EMEP4UK NOX and VOC fields. However, the only way around this would be joint bias-correction of NOX and VOCs which is beyond the scope of current research, and certainly beyond the scope of the current paper.

Lastly, we note that an alternative approach to would be to use a ML algorithm to directly predict the bias, i.e. the response would be observation{s,t} - EMEP4UK{s,t} where s and t denote location and time respectively. Ozone predictions would be obtained by predicting the bias and then using this to correct the EMEP4UK ozone prediction. Whilst this approach does, in a sense, treat the EMEP4UK ozone outputs as a response, it does not take into consideration the strength of the relationships between observations and EMEP4UK output.

**We have added the following text to Section 4.3 of the revised manuscript:**

"The ozone-NOX-VOC response is dealt with by EMEP4UK, before EMEP4UK ozone is used in our ML model and in all cases (i.e., base run + scenario runs). Consequently, our predicted surfaces may inherit some inaccuracies from the EMEP4UK surfaces resulting from the representation of ozone regimes in the CTM. However, we note that the model is widely used and well evaluated and has been used in this application (i.e., NOX sensitivity analysis) previously (Vieno et al., 2010)."

I was intrigued by Fig. 3. From how I look at what it says, the SHAP value is typically negative for the response to the EMEP model. Doesn't this mean that most of the time, the GB model responds negatively? Does this not mean that the GB model response should be in the opposite direction? (It might be more precisely explained). Even if it only means that the response is muted, what is the physical/chemical reasoning for such in terms of actually believing the model response to controls?

SHAP values should not be interpreted as classical regression coefficients or residuals, as they do not show directly the magnitude or functional form of the relationships between the features and the response. Instead, SHAP values display the difference between the average (expected) value of the response and the conditional average (expectation) of the response given a specific value of the feature. It follows that a linear regression with positive coefficient will have both negative (for feature values which predict below average response values) and positive (for feature values which predict above average responses) SHAP values.

Consequently, the negative SHAP values show that inputting low EMEP4UK ozone results in the model predictions being lower than the average ozone value, with higher-than-average predictions from high EMEP4UK ozone input. Further we see a clear monotonic increase in SHAP values as EMEP4UK ozone increases; this is consistent with model predictions increasing monotonically with EMEP4UK ozone values. We conclude that the GB model is responding 'positively'. A similar pattern is seen with other features, e.g., daily maximum T2 for which negative SHAP values occur when the temperature is lower and vice versa for positive SHAP values. We therefore conclude that low (high) temperatures predict below (above) average ozone values.

Lastly, we note that, of all the features, EMEP4UK ozone has the greatest range in magnitudes of SHAP values; this implies that EMEP4UK ozone is the most important of all features in explaining variability in the response.

This brings up the second potential question: why not include emissions in the GB model? This can help assure that the "package" (i.e., the GB model) is capturing the response over time, assuming that it correctly captures the response.

We appreciate the reviewer's question regarding the incorporation of emissions into the GB model. While we acknowledge the significance of emissions in capturing temporal responses, we have chosen not to include EMEP4UK  $NO_X$  emissions within the GB model to avoid introducing additional bias to the predictor features. Integrating EMEP4UK  $NO_X$  emissions into the model would have required subjecting them to the same downscaling treatment as the EMEP4UK ozone data.

In lieu of directly incorporating EMEP4UK NOX emissions, we have used distance to various road types to approximate the NOX characteristics of each location. While this approach indirectly encompasses the influence of NOX, a comprehensive treatment of NOX within the GB model is beyond the current scope of our study. Additionally, the presence of sharp gradients in NOX emissions introduces a potential risk of introducing spurious features during the downscaling process. Managing such complexities would have posed nontrivial challenges and extended beyond the intended focus of this paper.

Our analysis in the final part of the paper is oriented towards providing a simplified sensitivity test to offer a preliminary assessment of the ozone response to simplified  $NO_X$  emission reductions. We believe that the chosen framework is adequate for achieving this purpose and contributes valuable insights within the confines of our study's scope.

Their model evaluation is not well described. Are the R2 given for how well it captures peak ozone daily at each site for each year? (Is it for all sites, all days?) It seems so, and this should be emphasized. It seemed like it could also be for a different metric spatially. The table caption should be very precise as to what is shown.

Yes, all reported R2 results are calculated across all sites and all days. We have updated the Table 2 and Table 3 captions to make this clearer by adding "All metrics are calculated using all available data i.e., daily data across all sites.".

Also, I will note, they have NOT verified the model accuracy. They have estimated it. Indeed, they should describe what they mean by verification and what metrics and cut-offs are used to verify the model. As noted by Oreskes et al., (Science, ~1993) environmental models can not be verified. I think in this case, they mean evaluated.

**We have removed the word "verified" from line 275 and have replace it with the word "evaluated".**

I was also intrigued by why their 70/30 results are less good than the 10-fold CV. Also, it seemed for an ML model, the correlations might be a bit low: more discussion is needed.

The reviewer may have misread the results from the table. The 70/30 train/test results are better than the 10-fold CV results – average R2 results for 70/30 test is 0.80 vs 0.70 for the 10-fold CV. Similarly, the average RMSE is lower for the 70/30 test than the 10-fold CV at 10.61  $\mu$ g/m3 vs 13.07  $\mu$ g/m3, respectively.

We could tune the machine learning model to produce a high correlation with measurement data however, this would result in an over-fitted model incapable of spatial extrapolation which is necessary due to the sparseness of the measurement data. We therefore tried to balance the performance of the model fitted to seen training with the ability to predict unseen data.

Our results are not inconsistent with other machine learning downscaling approaches for ozone. Liu (2020) applied a similar method to produce a spatiotemporal surface of ozone concentrations in China from 2005 to 2017 and achieved a daily site cross-validation R2 score of 0.64 and RMSE of 27.27  $\mu$ g/m3. Ren (2020) investigated various machine learning models to predict ozone across the US and the highest spatial validation R2 score was 0.68.

In the end, I am not sure they can overcome the main concern, i.e., using the model to assess response to controls when the approach used is not evaluated and may actually introduce a bias in the response, potentially even having the wrong sign. This needs to be further assessed before publication.

We acknowledge the reviewer's main concern over using the model to assess response to  $NO_X$  controls. However, as discussed in our response to the first comment made by the reviewer, our objective is not to quantify the response of ozone to control variables but to provide a downscaled higher resolution of the EMEP4UK ozone surfaces produced under both current and alternative  $NO_X$  scenarios. We again emphasise that the ozone-NOX-VOC

response is dealt with by EMEP4UK before any downscaling occurs. It is also clearly well beyond the scope of this study to perform a detailed model evaluation of that complex response, though previous studies (e.g., Vieno et al., 2010) have examined some aspects.

The downscaled surfaces are produced using the assumption that the functional relationships inferred from in-situ measurements and output from EMEP4UK run under current atmospheric conditions, still hold for (a) locations for which there are no in-situ measurements and (b) EMEP4UK ozone surfaces produced under alternative NOX scenarios. It is true that we cannot assess the validity of this assumption, but this criticism can be levelled at all downscaling methodologies – it is not a limitation that is unique to our work. Lastly, the controls are not used in the downscaling model, and so for our purposes it should not matter if the downscaling predicts ozone concentrations that are inconsistent with the raw EMEP4UK NOX-VOC output.

**Reviewer 2**

The manuscript by Gouldsbrough et al. makes use of surface ozone simulations from the EMEP4UK model and uses a machine learning approach to downscale this data to 1 km x 1km, thus providing a higher resolution dataset for assessing O3air quality but also remove a positive bias in the model. This is achieved by exploiting surface ozone data from the AURN. Overall, the manuscript is well written and provides an interesting insight of how to fuse model data and observations together to provide a supposedly more robust product for air quality assessments. [My comments/suggestions are as follows:]

**We thank the reviewer for their comments and suggested revisions to our manuscript. Responses to individual points are listed below.**

Major Comments:

1. In the machine learning approach, you use surface  $O_3$  observations but could you also exploit information on  $NO_2$  and aerosols from AURN.  $O_3$  is influenced by concentrations of both, so would having surface data to constrain the biases you have in precursor pollutants help with the  $O_3$  downscaling? Also, would other meteorological variables (e.g. model cloud cover and photolysis) and surface type variables (e.g. vegetation cover and roughness – for deposition) help constrain the  $O_3$  downscaling?

We appreciate the reviewer's suggestion regarding the potential use of additional data sources, such as NO2 and aerosol measurements from the AURN, as well as the inclusion of other meteorological and surface type variables, to enhance the ozone downscaling in our machine learning approach. We recognize that incorporating these additional features could potentially provide valuable information for further constraining the ozone downscaling model. Considering the reviewer's suggestion, we plan to explore the inclusion of these additional variables in future research to investigate their impact on improving the ozone downscaling an expanded dataset that includes the suggested variables. However, given the scope of the current study, we regret that we are unable to incorporate these variables in the analysis at this time.

2. I'm not an expert in machine learning but as the AURN O3 data is used in the downscaling approach, would an independent data set be more appropriate for assessing the skill of it? From what I understand, you use the "training data" from the AURN data (i.e. a subselection) and then apply the method to all the data to generate the final product. However, if you then use the same data to evaluate the product, I'm not sure this can be deemed "independent" and a suitable evaluation of the product. Would it be worthwhile comparing the final product and EMEP4UK with an independent O3 observational data set to see if this downscaling approach truly works. E.g. ozonesonde. Or are there separate EMEP sites not included in the AURN data?

We appreciate the reviewer's point regarding the evaluation of our downscaling approach using the same dataset for training and evaluation. We agree that independent validation is crucial for assessing the model's performance. We used a 10-fold site cross-validation strategy. In this strategy, we divided the AURN dataset into ten subsets. The model was trained on nine subsets, while the remaining subset served as the evaluation set. We repeated this process ten times, ensuring that each subset was used for evaluation exactly once. By adopting this approach, we aimed to simulate the model's performance at locations where it has not encountered information during training.

To ensure the model's ability to generalize and avoid overfitting, we carefully tuned the model to strike a balance between model complexity and generalization, allowing the model to capture underlying relationships between the input variables and the target ozone variable effectively. This approach ensures that the model learns representative patterns and associations from the training data while avoiding over-reliance on specific instances or noise in the dataset.

Although an entirely independent ozone observational dataset might provide further validation, we believe that our cross-validation approach, coupled with appropriate model tuning, has enabled the model to extrapolate its learnings to unseen locations. Besides, we are unaware of any long-term independent surface ozone data over the UK that could serve this purpose.

3. The average and trends values are calculated for EMEP4UK, the downscaled product and the observations on a regional basis. However, I'm concerned that you are potentially getting regional domain statistics for the modelled data and then comparing with observations but there is a substantial representation bias (i.e. lots more model pixels then surface observational sites). Therefore, in your supporting information document, I think you need to see if the model/downscaled metrics are sensitive if you only sample the model/downscaled data like the observations.

We appreciate the reviewer's attention to this aspect of our analysis. Indeed, when comparing regional domain statistics with observations, it's expected that a reduced sample size, as seen in observations, could yield different trend and average statistics. However, one of the key conclusions drawn from our study is that relying solely on a limited number of point locations may lead to a different understanding of regional behaviour compared to the more comprehensive and robust perspective provided by considering the entire region.

For the reviewer's benefit, we did consider how the statistics would differ when using smaller measurement location samples for each of the regional model results. We calculated the yearly mean trends using all cells within a region and compared to selecting only nearest cell locations to the measurement stations (see

Table *I* below). The metrics are indeed sensitive to selecting few sparse samples in both the downscaled and original EMEP4UK surfaces, as might be expected.

We have therefore added the following text "and therefore more comprehensive regional estimates" to line 435 of the manuscript. However, we do not wish to add more tables and results to the paper to avoid lengthening the paper further and diluting our main message. We note again that our best estimates are from the cell-wise product as opposed to the sparse measurement data.

Table 1: Annual mean trend of MDA8 ozone  $(\mu g/m^3/yr)$  per region for 2001–2018, with 95% confidence intervals of the mean estimate shown in brackets. Downscaled and EMEPUK estimates are from nearest cell locations to measurement stations only. Significant trends are in bold.

| Region                   | Downscaled           | EMEP4UK             | Measurements        |
|--------------------------|----------------------|---------------------|---------------------|
| East Midlands (England)  | 0.04 [-0.30, 0.37]   | 0.33 [0.10, 0.56]   | 0.24 [-0.08, 0.55]  |
| East of England          | -0.16 [-0.47, 0.14]  | 0.15 [-0.09, 0.39]  | 0.09 [-0.19, 0.36]  |
| London                   | -0.11 [-0.42, 0.20]  | 0.48 [0.25, 0.72]   | -0.23 [-0.50, 0.05] |
| North East (England)     | -0.43 [-0.68, -0.17] | 0.11 [-0.11, 0.33]  | -0.18 [-0.42, 0.06] |
| North West (England)     | 0.13 [-0.13, 0.40]   | 0.39 [0.18, 0.60]   | 0.29 [-0.02, 0.59]  |
| Northern Ireland         | -0.03 [-0.23, 0.16]  | -0.06 [-0.26, 0.14] | -0.06 [-0.27, 0.15] |
| Scotland                 | -0.24 [-0.47, -0.02] | 0.20 [-0.03, 0.42]  | 0.18 [-0.04, 0.39]  |
| South East (England)     | -0.11 [-0.39, 0.17]  | 0.12 [-0.09, 0.35]  | 0.06 [-0.24, 0.37]  |
| South West (England)     | -0.22 [-0.46, 0.04]  | 0.00 [-0.21, 0.22]  | 0.19 [-0.10, 0.47]  |
| Wales                    | -0.21 [-0.46, 0.04]  | 0.21 [-0.01, 0.42]  | -0.13 [-0.36, 0.11] |
| West Midlands (England)  | 0.06 [-0.23, 0.35]   | -0.66 [0.45, 0.87]  | 0.28 [-0.01, 0.56]  |
| Yorkshire and The Humber | 0.09 [-0.14, 0.33]   | 0.38 [0.21, 0.57]   | 0.33 [0.02, 0.64]   |

4. Section 4.1: I do not fully understand the benefit of this section. Does 5-years really provide you with enough information on the inter-annual variability? You look at trends over the full time period, so you might as well look at the full data set? Or at least include the first 5-year period and compare that with the 2014-2018 period. That would let you assess inter-annual variability at the start and end of the record. Then use the trend analysis to look at long-term changes (i.e. rates) with time.

The objective of Section 4.1 is to offer a recent average representation of UK ozone behaviour. Recognizing the significant impact of inter-annual variability, particularly for metrics like yearly exceedance counts, we use a 5-year average to strike a balance between capturing these variations and providing a stable depiction of the data. This approach offers a reasonable assessment of the variables under scrutiny while mitigating some of the inherent year-to-year fluctuations. It is worth also acknowledging the substantial content already within the main text, which includes numerous figures, and an additional 15 figures and 8 tables within the appendix. Given the number of metrics potentially involved, the analysis suggested by the Reviewer would add substantial further length, and dilute the focus, while not being required to support our main conclusions. Minor Comments: Line 74: Delete ozone at end of line.

**The duplicate "ozone" has been removed.**

Line 100: Add , after "In Section 4".

**A comma has been added on line 100 after "In Section 4".**

Line 141: Should that be 2001-2017?

Yes, it should be 2001-2017. The Weather Research and Forecast (WRF) model version 3.7.1 was used in generating EMEP4UK data for 2001-2017, and WRF4.1 was used to generate EMEP4UK ozone for 2018.

Line 153: Would it be better to compare with NAEI emission data for e.g. 2019 and not 2020 as the latter was influenced by COVID19 and probably over-exaggerates the true emission decrease.

We thank the reviewer for this suggestion and have updated the text to instead include the 1970–2019 emission reduction of 73% in lines 154-155 of the updated manuscript.

Line 175: Distance to road acts as a proxy for NO2 concentration. Is this a linear relationship or treated as non-linear in your work as I suspect there would be a sharp drop off with distance in NO2 concentration?

All relationships between measurement ozone and input feature (including distance to road) are treated as non-linear due to the nature of the tree-based machine learning model used.

Line 178: Is this definition of London based on a subjective choice?

Yes, this is a subjective choice. A bounding box was a simple way of defining the London area.

Line 185: Would it not make sense to get data for Northern Ireland as well since you have got extra data sets for the other nations of the UK?

We are not aware of additional ozone monitoring networks for Northern Ireland that are not already included within the AURN network.

Line 188: Is 3-years a long enough record for a surface site to be included in the analysis. Since you are using this data to produce a data set for trend analysis, I would suspect 5-year at least but 10-years would be better.

We recognise that a longer record length would allow for better representation of a location to be incorporating within the model. However, restricting site inclusion to 5 or more years results in far less sites being able to be included within the model. We therefore chose a 3-year minimum to balance spatial density of sites with temporal representation of a location.

Lines 208-209: What quantitative approach do you use to implement Steps 3 and 4? I.e. what is used to determine no improved predictive skill in the approach?

The metrics from the validation experiments, both the 70/30 train/test experiment and 10fold CV, were used to determine when the model was sufficiently tuned. Further increases in performance for the 70/30 train/test experiment resulted in poor performance in the 10-fold CV test, as the model was overfitting to the training data.

**Line 239: What do you mean by "long-tail"? Not normally distributed?**

Yes, long tailed data refers to a distribution of data where large values occur infrequently, and the data is therefore not normally distributed. We have added "*i.e., non-normally distributed data with few but large extreme values*" to the sentence to expand on this within the manuscript.

Line 401: Defra 2021b reference style needs changing.

**We have removed the parenthesis around this reference.**

Figure 9 and Lines 495-497: I do not see the correlations between temperature anomaly and number of days where data > 100 ug/m3. Please quantify this relationship. Also, you are comparing number of days per heatwave with temperature anomaly. How long were the heat waves? Sample sizes suitable in time to get a relationship between  $\Delta T$  and N Days??

We agree with the reviewer that the Figure 9 did not show a clear correlation between temperature anomaly and number of days exceeding  $100 \,\mu\text{g/m}^3$ . We have updated the figure to show the number of days exceeding 25°C instead of the temperature anomaly and have included the following within the text: "number of days per year with daily maximum temperature exceeding 25°C, the minimum heatwave temperature threshold for the UK (McCarthy et al., 2019)". We believe this better shows the temperature-ozone exceedance relationships for the given years.

Figure A3 and similar: Instead of plotting all the data points for each year, would it be better to show the percentile values (e.g. 10, 25, 50, 75 and 90%) as this might be clearer and tell you more about the distribution for that year. With all data points plotted, some of the detail is difficult to resolve by the naked eye.

**We agree with the reviewer's suggested to show more meaningful summaries of each year's data. We have updated Figures A3 – A14. The figures now show the yearly boxplots to better demonstrate the distribution of data for each year.**

Figure A5: The AURN data tends to be more variable per year than the model data and downscaled data. Why is this and would this have an important impact on your downscaling approach. From what I can see, the downscaled data is struggling to capture the full variability in the observations? Especially for London.

We acknowledge that Figure A5 did not show the same level of variability per year in the downscaled surface compared to the measurement data, however we believe this to be due to the style of plot as the data points are too small and transparent to see in the downscaled figure compared with the larger and more opaque points in the measurement figure. We have updated Figure A5 to show the yearly distributions of data using boxplots, better demonstrating the variability within the downscaled data.

**References**

Liu, R., Ma, Z., Liu, Y., Shao, Y., Zhao, W. and Bi, J., 2020. Spatiotemporal distributions of surface ozone levels in China from 2005 to 2017: A machine learning approach. *Environment international*, *142*, p.105823.

Ren, X., Mi, Z. and Georgopoulos, P.G., 2020. Comparison of Machine Learning and Land Use Regression for fine scale spatiotemporal estimation of ambient air pollution: Modeling ozone concentrations across the contiguous United States. *Environment international*, 142, p.105827.

Vieno, M., Dore, A. J., Stevenson, D. S., Doherty, R., Heal, M. R., Reis, S., Hallsworth, S., Tarrason, L., Wind, P., Fowler, D., Simpson, D., and Sutton, M. A.: Modelling surface ozone during the 2003 heat-wave in the UK, Atmos. Chem. Phys., 10, 7963–7978, https://doi.org/10.5194/acp-10-7963-2010, 2010.

McCarthy, M., Armstrong, L., Armstrong, N.: A new heatwave definition for the UK. Weather. 74, 11. https://doi.org/10.1002/wea.3629, 2019.

---

## Editor Decision (ED1)

**Editor comment on Gouldsbrough et al.: *A machine learning approach to downscale EMEP4UK: analysis of UK ozone variability and trends**

Thanks for submitting a revision of your manuscript in response to the two reviews. As both reviewers had major concerns about the study, I had asked them for a second review. Please respond to the comments by Reviewer#2 on the revised version of the manuscript.

While the reviewer remains sceptic about your study, I think that it could stimulate further discussion and could be a starting point for further studies. Therefore, I favour publication in ACP. However, I agree with reviwer#2 that when revising the manuscript you omitted a lot of the information from the response to the reviewer comments. I understand that you are concern about the length of the manuscript and I agree, that it should not become much longer, but at the same time I feel that some additional information might be useful for the readers.

Therefore, I ask you to prepare a new minor revision of the manuscript taking into account the new review and the following aspects.

**Remarks on Response to Reviewer #1 (no 2nd review received)**

- Add the explanation on the interpretation of SHAP value to the manuscript. I think is necessary as SHAP values are not a widely known metric.

- Add a statement about not including emissions to text (3rd comment by Reviewer #1).

- The response argues with a citation of Ren et al. (2020); this reference along with the argument of Liu et al and Ren et al achieving similar correlation values should be added to the text.

**Remarks on Response to Reviewer #2 (numbers referring to major comments from reviewer's first review):**

1. ok

2. The text in section 3.1. in version 5 of the manuscript remains less detailed than the response to the reviewer as subsection 3.1 remained largely unmodified – please change and add details to the manuscript.

3. Please add the Table from your response to the appendix and summarise the results briefly in the main text also addressing the reviewer's concerns about the representativeness of model.

4. ok

---

## Author Response (AR2)

**Response to reviewer 2 comments**

Reviewer comments are in red, and author responses are in blue.

Firstly, the authors have undertaken a detailed response to the review comments (from both reviewers). However, while they provide useful information justifying their methods and results, they infrequently make updates/clarifications to the manuscript to make this clear. If the reviewers find something unclear in the manuscript, it is very useful for the authors to provide good, detailed responses, but the manuscript needs updating to address this as future readers may have the same/similar questions.

We thank the reviewer for their additional comments. We agree that some further updates around clarity would be beneficial and have therefore updated the manuscript to address the comments below and those of the Editor.

Secondly, I am concerned by the new Table 1 in response to my comment #3. The authors clearly show that the downscaling approach improves the model (e.g. in terms of the absolute values), however, this is not the case for the surface ozone trends. In my comment #3, I wondered how important the sampling differences were between the model and observations (i.e. for the regional statistics, I believe the model/downscaling uses all regional pixels while only a few observation sites are used). To try and address this, the authors provided Table 1 which shows the trends for the model/downscaled data sub-sampled to the observations (i.e. closest pixel/grid box). In the manuscript, the authors discuss trends from the three data sources (Figures 7 and 8) but now I am not convinced they are comparable given the model/downscaled data is essentially representing a different quantity to that of the observations. From their Table 1 in the response document, there is a mixed response in whether the co-located model or downscaled trends more closely match the observations. I agree the observations have less spatial coverage than that of the model/downscaled data, so are not truly representative of county regional trends. However, if the observations are used in the ML approach to generate an improved higher resolution surface ozone dataset, then I would expect the downscaled data, when co-located to the observations, to be more representative of what the observations are showing (if only for several sites used to determine an observational trend). As a result, can the authors be confident in their downscaling approach to produce a higher spatial resolution product of surface ozone to investigate temporal evolution? I feel this needs to be addressed in detail in the main manuscript.

We agree with the reviewer that additional text on this was needed in the manuscript. We have therefore added the following text to section 4.2 and have included the table mentioned in the appendix.
"The analysis presented above provides valuable insights into the trends derived from downscaled and EMEP4UK data across a given domain. The downscaled and EMEP4UK trends encompass all pixels within a designated area. To delve deeper into the sensitivity of the trend analysis concerning sample size, we undertook a sub-sampling process for both the downscaled and EMEP4UK data specifically at measurement locations. The resulting annual mean trends are

given in Table A9, demonstrating the impact of sample size on trend outcomes. However, a note of caution is warranted against drawing excessive conclusions from small, largely non-significant trends observed across datasets. Both the downscaled and EMEP4UK products are susceptible to sampling errors due to the process of condensing a coarse grid model to specific point locations. As a result, over-interpreting these trends might lead to misleading assumptions. Therefore, the trends derived from the gridded products are anticipated to be the most regionally representative when considering the entire domain."

**Response to editor comments**

Editor comments are in red and author responses are in blue.

Thanks for submitting a revision of your manuscript in response to the two reviews. As both reviewers had major concerns about the study, I had asked them for a second review. Please respond to the comments by Reviewer#2 on the revised version of the manuscript.

While the reviewer remains sceptic about your study, I think that it could stimulate further discussion and could be a starting point for further studies. Therefore, I favour publication in ACP. However, I agree with reviwer#2 that when revising the manuscript you omitted a lot of the information from the response to the reviewer comments. I understand that you are concern about the length of the manuscript and I agree, that it should not become much longer, but at the same time I feel that some additional information might be useful for the readers.

We thank the Editor for their encouraging remarks and have addressed all comments by adding additional text in the manuscript, as outlined below.

Therefore, I ask you to prepare a new minor revision of the manuscript taking into account the new review and the following aspects.

**Remarks on Response to Reviewer #1 (no 2nd review received)**

- Add the explanation on the interpretation of SHAP value to the manuscript. I think is necessary as SHAP values are not a widely known metric.

The following text has been added to Section 3.3 to further explain the interpretation of the SHAP values:
"Instead, SHAP values display the difference between the average value of the response and the conditional average of the response given a specific value of the feature. Positive SHAP values can co-occur with either high (red) or low (blue) values of a feature, and similarly for negative SHAP values."

- Add a statement about not including emissions to text (3rd comment by Reviewer #1).

The following text has been added to Section 2.3: "While this approach indirectly encompasses the influence of $NO_X$, a comprehensive treatment of $NO_X$ within the ML model is beyond the current scope of our study. Additionally, the presence of sharp gradients in $NO_X$ emissions introduces a potential risk of introducing spurious features during the downscaling process."

- The response argues with a citation of Ren et al. (2020); this reference along with the argument of Liu et al and Ren et al achieving similar correlation values should be added to the text.

The following text has been added to Section 3.2.2: "Our results are not inconsistent with other machine learning downscaling approaches for ozone. Liu et al., (2020) applied a similar method to produce a spatiotemporal surface of ozone concentrations in China from 2005 to 2017 and achieved a daily site cross-validation $R^2$ score of 0.64 and RMSE of 27.27 $\mu g/m^3$. Ren et al., (2020) investigated various machine learning models to predict ozone across the US and the highest spatial validation $R^2$ score was 0.68."

**Remarks on Response to Reviewer #2 (numbers referring to major comments from reviewer's first review):**

1. ok

2. The text in section 3.1. in version 5 of the manuscript remains less detailed than the response to the reviewer as subsection 3.1 remained largely unmodified – please change and add details to the manuscript.

The additional text below detailing the 10-fold cross validation test has been added to Section 3.2.1:

"To do this, we divided the measurement data into ten subsets by their location. The model was trained on nine subsets, while the remaining subset served as the evaluation set. We repeated this process for all subsets, ensuring that each subset was used for evaluation exactly once."

3. Please add the Table from your response to the appendix and summarise the results briefly in the main text also addressing the reviewer's concerns about the representativeness of model.

This table has been added to the appendix (Table A9), and additional text has been added to section 4.2 discussing the representativeness of the model.

4. ok